# Strong impacts on aerosol indirect effects from historical oxidant changes

Inger Helene Hafsahl Karset[1], Terje Koren Berntsen[1,2], Trude Storelvmo[1], Kari Alterskjær[2], Alf Grini[4], Dirk Olivié[4], Alf Kirkevåg[4], Øyvind Seland[4], Trond Iversen[4], and Michael Schulz[4]

[1]University of Oslo, Department of Geosciences, Section for Meteorology and Oceanography
[2]CICERO Center for International Climate Research, Oslo, Norway
[4]Norwegian Meteorological Institute, Oslo, Norway

*Correspondence to:* Inger Helene Hafsahl Karset (i.h.h.karset@geo.uio.no)

**Abstract.** Uncertainties in effective radiative forcings through aerosol-cloud interactions (ERF$_{aci}$, also called aerosol indirect effects) contribute strongly to the uncertainty in the total preindustrial-to-present-day anthropogenic forcing. Some forcing-estimates of the total aerosol indirect effect are so negative that they even offset the greenhouse gas forcing. This study highlights the role of oxidants in modeling of preindustrial-to-present-day aerosol indirect effects. We argue that the aerosol
precursor gases should be exposed to oxidants of its era to get a more correct representation of secondary aerosol formation. Our model simulations show that the total aerosol indirect effect changes from -1.32 Wm$^{-2}$ to -1.07 Wm$^{-2}$ when the precursor gases in the preindustrial simulation are exposed to preindustrial instead of present-day oxidants. This happens because of a brightening of the clouds in the preindustrial simulation, mainly due to large changes in the nitrate radical (NO$_3$). The weaker oxidative power of the preindustrial atmosphere extends the lifetime of the precursor gases, enabling them to be transported
higher up in the atmosphere and towards more remote areas where the susceptibility of the cloud albedo to aerosol changes is high. The oxidation changes also shift the importance of different chemical reactions and produce more condensate, thus increasing the size of the aerosols and making it easier for them to activate as cloud condensation nuclei.

## 1 Introduction

It is well established that changes in atmospheric aerosol abundance since preindustrial times have had a strong, albeit uncertain,
influence on Earth's climate over the last century. Atmospheric aerosols are not just impacting climate by directly absorbing and reflecting radiation, but also indirectly by acting as cloud condensation nuclei (CCN) and ice nuclei (IN). Through cloud albedo increases mediated by enhancements of CCN, aerosols brighten the clouds and enhance their cooling effect by increasing the reflection of incoming solar radiation (Twomey, 1977). More numerous cloud droplets may also alter rain formation mechanisms, thus the cooling effect could be further enhanced by suppressed precipitation followed by increased cloud life-
time, cloud amount and cloud extent (Albrecht, 1989; Pincus and Baker, 1994). The impact of IN changes remains uncertain (Storelvmo, 2017; Lohmann, 2017).

Aerosol indirect effects on Earth's radiation budget are often quantified in terms of their effective radiative forcing (Myhre et al., 2013). Unlike instantaneous radiative forcing, effective radiative forcing includes effects from rapid tropospheric ad-

justments (Boucher et al., 2013). Otherwise, it does not include any feedbacks in the climate system. Model studies of direct and indirect effects typically carry out two simulations, with aerosols and aerosol precursor gases from preindustrial times (PI) and present-day (PD), respectively. The difference in cloud forcing, measured as effective radiative forcing between the two simulations, represents the total aerosol indirect effect if the direct aerosol effect in cloudy skies is negligible (Ghan, 2013).

Results from several model studies show that this number varies considerably. To what extent aerosol-cloud interactions have contributed to the global radiative forcing in the anthropocene remains highly uncertain and continues to be a research topic of much interest. Lohmann (2017) shows that model estimates of $ERF_{ari+aci}$ (ari: aerosol-radiation interactions, aci: aerosol-cloud interactions) vary from -0.07 to -3.41 $Wm^{-2}$, while the fifth Assessment Report (AR5) from the Intergovernmental Panel on Climate Change (IPCC) gives an expert judgement of $ERF_{ari+aci}$ of -0.9 $Wm^{-2}$, with a 5 to 95 % uncertainty range of -1.9

to -0.1 $Wm^{-2}$ mostly coming from the uncertainties in the aci-component (Boucher et al., 2013). Uncertainties in the natural background emissions have been highlighted as a large contributor to the uncertainty in the indirect effects (Lohmann et al., 2000; Kirkevåg et al., 2008; Hoose et al., 2009; Carslaw et al., 2013), while Gettelman (2015) pointed out that its sensitivity to parameterizations of microphysical processes in global models is even higher. In this study, we examine a third factor, namely the oxidants involved in the formation of aerosols.

Aerosols may enter the atmosphere directly, or they can be formed after in situ oxidation of precursor gases to condensable species (Seinfeld and Pandis, 2016). The oxidation process yields secondary gases with lower saturation vapor pressure, which allows them to either condense on already existing particles or nucleate into new particles under atmospheric conditions. Both processes depend on the amount of emitted precursor gases, but also on the atmospheric oxidation capacity. While model studies of PD-PI aerosol indirect effects usually point out that they use different emissions of aerosols and aerosol precursor

gases for the two different time periods, the choice of oxidant levels is usually not specified (Lohmann and Diehl, 2006; Menon and Rotstayn, 2006; Hoose et al., 2008; Storelvmo et al., 2008; Lohmann, 2008; Lohmann and Ferrachat, 2010; Wang et al., 2011; Yun and Penner, 2013; Neubauer et al., 2014; Gettelman, 2015; Gettelman et al., 2015; Tonttila et al., 2015; Sant et al., 2015). A notable exception is Salzmann et al. (2010), who use different oxidant levels for the different eras. Personal communication with scientists from different modeling groups confirms that it is common to use PD-oxidants for both PD- and

PI-simulations (U. Lohmann, C. Hoose, A. Kirkevåg, A. Gettelman, D. Neubauer, personal communication, 2017).

   Human activity has influenced the oxidant level mainly through increased emissions of CO, $NO_x$, and $CH_4$ from fossil fuel combustion, biomass burning and the use of fertilizers in agriculture (Crutzen and Lelieveld, 2001). Due to this anthropogenic activity, precursor gases emitted into the PI-atmosphere were exposed to a different oxidant level than the gases emitted today, implying a difference in the rate and distribution of new particle formation in the atmosphere. The aim of this study is to

30 quantify this difference and to give a more realistic estimate of the total PD-PI aerosol indirect effect by letting the precursor gases in the PI-simulation (the simulation with emissions of aerosols and aerosol precursor gases from PI) be exposed to an oxidant level that is representative for its era.

   Due to counteracting effects, the sign and magnitude of the global mean historical oxidant change is uncertain (Naik et al., 2013a, b; Murray et al., 2014). While in a low $NO_x$-regime, CO and $CH_4$ act as sinks for the hydroxyl radical (OH), one

of the most important oxidants in the troposphere, the opposite is the case in a high $NO_x$-regime (Collins et al., 2002). As a

consequence, OH has experienced an increase in polluted areas where the $NO_x$ level is high, while it has decreased in remote areas where the $NO_x$-level is low and the $CH_4$ level is high due to their different lifetimes (Wang and Jacob, 1998; Prinn, 2003). The situation is different for ozone ($O_3$), where an increase in $NO_x$, CO or $CH_4$ usually favours $O_3$-production in both low and high $NO_x$ regimes (Seinfeld, 1989; Chameides et al., 1992). This also holds for the $NO_3$ radical, which is produced through reactions between $NO_x$ and $O_3$ (Wayne et al., 1991) and probably was present at lower levels everywhere in preindustrial times.

Difficulties in measuring the oxidants directly from the atmosphere and the lack of information about oxidants in sediments and ice cores has resulted in limited information about the atmospheric oxidant level (Pavelin et al., 1999). This is especially the case for the time period before the industrial era, were it is limited to simple measurements of surface ozone from a few European stations (Volz and Kley, 1988). Despite this limitation, results from model simulations based on information about emission changes, in combination with the few oxidant measurements that exist, give an indication of how the oxidative power of the atmosphere has changed since preindustrial time (Prinn, 2003; Berntsen et al., 1997; Wang and Jacob, 1998; Tsigaridis et al., 2006; Naik et al., 2013a, b; Young et al., 2013; Murray et al., 2014; Khan et al., 2015).

When trying to get a better understanding of the response of clouds to aerosol perturbations, or when comparing this effect between models, the choice of oxidant level may not be important as long as there is consistency between the different models. However, the oxidant level may be important when the modeled preindustrial-to-present-day total aerosol indirect effect is used as an estimate of the contribution from aerosol-cloud interactions to the total forcing of climate change since PI, as was done in IPCC AR5. Recent global model estimates of the aerosol indirect effects do, to a larger extent than before, represent more of the gas to aerosol formation processes through oxidation followed by nucleation (Boucher et al., 2013; Lohmann, 2017), increasing the importance of understanding the effects and the model treatment of the oxidants. More and more models will also incorporate an interactive atmospheric gas phase chemistry in transient climate studies, making the characterisation of effective radiative forcing a larger challenge. With this study we aim to use model simulations to investigate the impact on aerosol indirect effects from historical oxidants changes by letting the aerosol precursor gases in the PI-simulation be exposed to PI- instead of PD-oxidant level.

Information about the model and the configurations applied in this study is found in Sect. 2. The experimental setup for the default model configuration and the experimental setups where the impact of separate oxidant changes is found in Sect. 3. In Sect. 4, the results are presented and discussed, divided into subsections focusing on the effect of the oxidant changes on the aerosol number concentration (Sect. 4.1.1), on the cloud droplet number concentration (Sect. 4.1.2) and on the aerosol indirect effect (Sect. 4.1.3). The results and discussions of the sensitivity tests where the oxidant changes were separated are found in Sect. 4.2, while six other sensitivity tests are studied in Sect. 4.3.

## 2 Model

### 2.1 General description

The model used in this study is CAM5.3-Oslo (Kirkevåg et al., 2018), which is an updated version of the atmospheric component of the Norwegian Earth System Model (NorESM) (Bentsen et al., 2013; Iversen et al., 2013; Kirkevåg et al., 2013).

CAM5.3-Oslo is based on the Community Atmospheric Model version 5.3 (Neale et al., 2012; Liu et al., 2016), but has its own aerosol module (OsloAero). It also includes other modifications, such as the implementation of heterogeneous ice nucleation (Wang et al., 2014; Hoose et al., 2010). OsloAero has 21 aerosol tracers, distributed among six species (sulfate ($SO_4$), secondary organic aerosol (SOA), black carbon, organic matter, mineral dust and sea-salt), four precursor gases ($SO_2$,

dimethyl sulfide (DMS), isoprene and monoterpene), three condensable gases (sulfuric acid ($H_2SO_4$), $SOA_{LV}$ and $SOA_{SV}$) and $H_2O_2$. DMS-emissions are wind-driven and based on Nightingale et al. (2000), emissions of $SO_2$ are interpolated from a pre-scribed monthly mean decadal climatology given by Lamarque et al. (2010). The emissions of $SO_2$ in CAM5.3-Oslo deviates from Lamarque et al. (2010) when it comes to aircraft emissions and volcanic emissions, where the former is not included in CAM5.3-Oslo and the latter is included in the model but not in Lamarque et al. (2010). The emissions of the Biogenic Volatile

Organic Compounds (BVOCs) isoprene and monoterpene are calculated online every timestep of half an hour by a satellite phenology version of the Community Land Model version CLM4.5 (Oleson et al., 2013), using the Model of Emissions of Gases and Aerosols from Nature version 2.1 (MEGAN2.1) (Guenther et al., 2012), where the emissions are impacted by both radiation and temperature, inducing a diurnal variation. An overview of global emissions and burdens of the precursor gases in CAM5.3-Oslo is found in Table 1. The aerosol nucleation is based on Makkonen et al. (2014), with improvements described

in Kirkevåg et al. (2018). This nucleation scheme is divided into two parts, where the binary homogeneous sulfuric acid-water nucleation based on Vehkamäki et al. (2002) can act in the whole atmosphere, while the activation type nucleation of $H_2SO_4$ and organic vapor based on Eq. (19) in Paasonen et al. (2010) occurs only in the boundary layer. The survival rate of particles with diameter from 2 nm to 23.6 nm (where the upper limit corresponds to the smallest sized particles that are accounted for in the aerosol number concentration in the model) follows Lehtinen et al. (2007). The stratiform clouds are described by the

two-moment bulk microphysics scheme MG1.5, that is almost identical to MG1 described in Morrison and Gettelman (2008), but with cloud droplet activation moved before the cloud microphysical process rates calculations (Gettelman, 2015; Gettelman and Morrison, 2015).

Methods by Ghan (2013) are used for calculating the effective radiative forcing of aerosols. The part called "cloud radiative forcing", or $\Delta C_{clean}$ is often used as a measure of the total aerosol indirect effect, where it represents the difference in the top of

the atmosphere total cloud forcing between simulations performed with different aerosols. The "clean"-subscript indicates that the cloud forcing is based on separate calls to the radiation code where the scattering and absorption of radiation by the aerosols in the air around the cloud is neglected. $\Delta C_{clean}$ also includes semi-direct effects, but additional simulations with CAM5.3-Oslo with non-absorptive aerosols have shown that this term is negligible compared to the indirect effects in the model global mean PD-PI values (Kirkevåg et al., 2018). Henceforth we use $\Delta C_{clean}$ as a measure of the total aerosol indirect effect in this study.

**2.2   Oxidant chemistry**

CAM5.3-Oslo includes simple chemistry for sulfur and SOA species, which makes use of the chemical preprocessor MOZART (Emmons et al., 2010) modified for the CAM framework (Liu et al., 2012). The preprocessor is a numerical scheme that generates code to the model based on some input chemical reactions and rates. The generated code provides information of how the chemical tracers evolve as a function of concentration of chemical species. Reactions (R1-R10) in Table 2 represent

the gas phase oxidation of the precursor gases in the model. $SOA_{LV}$ and $SOA_{SV}$ are both gaseous SOA (SOA(g)), low volatile and semi-volatile respectively, where only 50 % of the former can take part in nucleation, while both can condense on already existing aerosols. While (R2) represents the H abstraction part of the complex reaction where DMS is oxidized by OH, (R3) represents the OH addition part. At standard conditions (temperature of 273.13 K and pressure of 1013 hPa), the ratio between

the reaction rates of (R2) and (R3) is 7/13 (R2/R3). Methanesulfonic acid (MSA) is produced in (R3) following Chin et al. (1996). Since CAM5.3-Oslo does not trace MSA, 20 % of the MSA is put into the $SOA_{LV}$-tracer, while 80 % is put into the $SOA_{SV}$-tracer. The exact yields are unknown, but there are studies supporting that MSA can obtain low enough volatility to contribute to new particle formation and growth (Bork et al., 2014; Willis et al., 2016; Chen and Finlayson-Pitts, 2017). The oxidation of BVOCs in (R5-R10) are based on Makkonen et al. (2014), but with some extensions explained by Kirkevåg et al.

(2018). The yield of 15 % for monoterpenes (considered to be $\alpha$-pinine in this model) is widely used in other global models (Dentener et al., 2006; Tsigaridis et al., 2014). The yield for isoprene varies more between different laboratory and model based studies ($0.9-12$ %) (Lee et al., 2006; Kroll et al., 2005; Spracklen et al., 2011; Jokinen et al., 2015), where the yield applied in CAM5.3-Oslo of 5 % is within this range.

     The model also includes aqueous phase oxidation of $SO_2$ by $H_2O_2$ and $O_3$ (Tie et al., 2001; Neale et al., 2012). $H_2O_2$

production and loss are calculated online through reactions (R11-R13) in Table 2.

     The concentrations of the other oxidants ($NO_3$, $O_3$, OH and $HO_2$) are prescribed by monthly mean values produced by the global full chemistry model CAM-chem v3.5 in the study of Lamarque et al. (2010). PD and PI-values used in this study are taken from decadal climatologies around year 2000 and 1855 respectively, and the percent change in the annual mean values can be seen in Fig. 1. $NO_3$ experiences a very large change between PI and PD (up to more than 1000 % in the northern

hemisphere), which is also seen in other model studies that show good agreement between modeled present-day concentrations of $NO_3$ and observations (Khan et al., 2015). The prescribed PI-values of surface layer $O_3$ in the region around Paris used in this study are around a factor two higher than the measured PI-values at a station near Paris in the study of Volz and Kley (1988) ($\sim$10 ppb). This overestimation of the PI-level of $O_3$ compared to observations corresponds with finding from other studies (Parrish et al., 2014). Evaluation of present-day concentrations of OH in a comparable version of CAM-chem shows reasonable

agreement with the Spivakovsky et al. (2000) climatology (Lamarque et al., 2012). Simulated tropospheric concentrations of $O_3$ also agree well with ozone sondes, except for an overestimation over Eastern US and Europe (Lamarque et al., 2012; Brown-Steiner et al., 2018).

     CAM5.3-Oslo applies a daily cycle to OH and $HO_2$, which is not included in CAM5.3. One should also be aware that the ozone climatology used for the radiation in the model is different from the ozone climatology used for the chemistry (the ozone

climatology for radiation is the same in the PI- and PD-simulations).

## 2.3 Configurations

The model was configured with a horizontal resolution of $0.9°$ (latitude) by $1.25°$ (longitude) and 30 hybrid levels between the surface and $\sim$3 hPa. The simulations were carried out using nudged meteorology produced by the model itself to constrain the natural variability (Kooperman et al., 2012). The horizontal wind components (U, V) were nudged with a relaxation time scale

of six hours, while the temperature was evolving free, allowing impacts by aerosol perturbations, which could be important when calculating indirect effects (Zhang et al., 2014). Prescribed climatological sea surface temperatures and sea-ice extent from the mean of 1982-2001 were used in all simulations, as well as greenhouse gas concentrations and land use information from year 2000.

## 3   Experimental setup

### 3.1   General

Figure 2 describes how the simulations were carried out. The model was first run for six years to generate instantaneous meteorological data using PD-conditions for emissions, prescribed oxidant, and all other boundary conditions. All other simulations were nudged to the meteorology of this simulation. For each modification to the default model setup, three different simulations were carried out. These three simulations used the prescribed precursor- and aerosol emissions and oxidant concentrations given in Table 3. Each of them was restarted from an earlier simulation that was already spun up for two years with free meteorology, applying emissions and oxidants from the same era. The nudged simulations where then run for four years, where the last three years were analyzed. Sensitivity tests with CAM5.3-Oslo (not shown here) show that analyzing only these three years gives a standard error due to natural variability of only 0.01 $\mathrm{Wm}^{-2}$ for the total aerosol indirect effect, and a magnitude of the total aerosol indirect effect that is the same as when running the nudged simulations for 11 years and analyzing the last ten years. To lower the computational cost, the simulations in this study apply the setup described above, except for one sensitivity test in Sect. 4.3 where longer simulations with free meteorology are examined. The first set of simulations used CAM5.3-Oslo as described in the previous section, without any other modifications to the code. We name these simulations ORG, and the impact of historical oxidant changes on the PD-PI total aerosol indirect effect in CAM5.3-Oslo are quantified by the difference we obtain (relative to the PD simulation PDAER_PDOXI_ORG) when switching between the two PI-simulations PIAER_PDOXI_ORG and PIAER_PIOXI_ORG.

### 3.2   Decomposing the oxidant change

To estimate the importance of the different changes in the individual oxidants between PI and PD, four additional simulations with PI-aerosols were carried out. In these simulations, the oxidant of interest was changed to PI-concentrations, while all other oxidants were kept at PD-levels. Acknowledging the complexity of oxidant chemistry, one can not expect that separate oxidant changes in separate simulations will add up to the same result as changing them all simultaneously. To explore the importance of this non-linearity, another four additional simulations were performed, keeping all oxidants from PI except for the one of interest, which was set to PD-levels.

## 4 Results and discussion

### 4.1 Original setup

The top panels of Fig. 3 show the PD-PI indirect effect for (a) shortwave radiation, (b) longwave radiation, and (c) total radiation when using the standard setup with PD-oxidants in both simulations. The bottom panels of Fig. 3 show the impact of historical oxidant changes on the PD-PI indirect effect. Figure 3(d) shows that letting the precursor gases in the PI-simulation be exposed to oxidants from its era, instead of oxidants from PD, makes the shortwave indirect effect 0.39 Wm$^{-2}$ less negative (changing from -1.48 Wm$^{-2}$ to -1.09 Wm$^{-2}$). This implies that the clouds in the PI-simulation with PI-oxidants are cooling the climate more through SW-effects than the clouds in the PI-simulation with PD-oxidants, reducing the difference in shortwave cloud forcing between PI and PD. Figure 3(e) shows that the change in longwave indirect effect is -0.14 Wm$^{-2}$ (from 0.16 Wm$^{-2}$ to 0.02 Wm$^{-2}$), meaning that the clouds in the PI-simulation with PI-oxidants are warming the climate more through increased absorption of longwave radiation, reducing the difference in longwave cloud forcing between PI and PD. Figure 3(f) shows a total (shortwave + longwave) change in the indirect effects of +0.25 Wm$^{-2}$ (changing from -1.32 Wm$^{-2}$ to -1.07 Wm$^{-2}$), meaning that the PI-clouds with PI-oxidants are cooling the climate more than the PI-clouds with PD-oxidants, thus making the indirect effect less negative. The largest changes in the shortwave indirect effect occur over ocean, especially over the North Pacific, off the west coast of America, in remote areas between 30° S and 60° S and over the Indian Ocean. The changes in the longwave indirect effect mainly take place in the polar regions and over the Indian Ocean.

Different cloud- and aerosol changes can help explain the resulting change in the indirect effect. Some of these are presented in Fig. 4. In the global mean, switching to PI-oxidants in the PI-simulation results in (a) more numerous aerosol particles (+9.2 %), (b) more numerous cloud droplets (CDNC) (+3.7 %), (c) smaller cloud droplets (-1.5 %), (d) larger cloud fraction (+0.26 %), which is mainly caused by changes in the low cloud fraction, and (e) larger total gridbox averaged liquid water path (LWP) (+1.7 %). The size of the cloud droplets in Fig. 4(c) is taken from the cloud top layer of the stratiform clouds.

The sign of the changes in the global mean cloud and radiative properties seen in Figs. 3 and 4 is as expected for an increase in the global mean aerosol number concentration. We will now further investigate why the oxidant changes enhance the aerosol number concentration. Figures 3 and 4 show that the distribution of the changes in aerosol number concentration does not always correspond directly to the distribution of the changes in the cloud and radiative properties. This indicates that it is not only the change in aerosol number concentration that is important for the result, but also changes in the composition of the aerosols and in the atmospheric conditions where the aerosol changes take place.

#### 4.1.1 The increase in aerosol number concentration

Since the formation of new aerosols depends on the availability of low-volatility gases, and the PI-atmosphere consisted of relatively small amounts of oxidants to produce secondary gases with reduced volatility, one could expect a reduction in the aerosol number concentration when switching from PD- to PI-oxidants. This is the opposite of what Fig. 4(a) shows. The increased lifetime of the precursor gases and the aerosols seen in Table 4 partly explains this. When the oxidizing power of the atmosphere is reduced, the precursor gases with high volatility are transported higher up in the atmosphere before they

are oxidized. This is seen in Fig. 5, where the relative change in chemical loss of (a) DMS, (b) $SO_2$, (c) isoprene, and (d) monoterpene through oxidation is negative close to the surface, but positive higher up in the atmosphere when switching from PD- to PI-oxidants in the PI-simulation. This pattern corresponds well with the change in the vertical profile of the aerosol number concentration seen in Fig. 9(a), with lower values close to the surface, but larger values above $\sim$900 hPa. Aerosols formed from gases higher up in the atmosphere are not removed by deposition as easily as aerosols formed closer to the surface (Jaenicke, 1980; Williams et al., 2002). This is seen in the results of this study where the dry deposition of the newly formed nucleation mode $SO_4$ and SOA decreases by 2.6 %. The wet deposition stays the same. This total decrease in deposition is one of the factors contributing to the increase in the aerosol number concentration seen in Fig. 4(a).

It is not only the vertical transport of the gases that changes. The reduced oxidation capacity also increases the horizontal transport of the primary precursors away from the source regions. This is, e.g., seen in Fig. 6 for DMS, the main precursor gas over ocean, where most of the aerosol-, cloud- and radiation changes occur. Figure 6(a) shows the distribution of DMS-emissions, which is equal in all PI-simulations, while Fig. 6(b) shows the change in the chemical loss of DMS through oxidation when switching from PD- to PI-oxidants. Increased horizontal transport happens from areas with negative values to areas with positive values, since chemical loss through oxidation is the only way DMS can be lost in the model. The increase is especially pronounced in the North Pacific, with increased transport further south and towards the Arctic, but is also found in the southern ocean with increased transport from the large emission sources close to the coast towards the remote ocean. Figure 6(c) shows that this transport results in increased aerosol formation close to the surface in areas that receive more DMS with PI-oxidants. Since the precursor gases are spread more in space with PI-oxidants, towards more remote areas where the background concentration of aerosols are low, the coagulation sink during the nucleation process is reduced, contributing to an increase in the formation rate. In CAM5.3-Oslo, "formation rate" describes the formation of aerosol particles with diameters of 23.6 nm, which is the size limit a particle must achieve to be accounted for in the aerosol number concentration (Figs. 4(a) and 6(e)). "Nucleation rate" describes the formation of aerosol particles with diameters of 2 nm. As for all aerosols, the particles between 2 and 23.6 nm can be lost through coagulation with background aerosols. Figure 6(d) shows how the coagulation sink of these particles changes when switching from PD- to PI-oxidants in the PI-simulation. The reduction in the coagulation sink is especially large close to the strong DMS-emissions sources (Fig. 6(d)). The areas over ocean with increased formation rate close to the surface correspond well with the areas in Fig. 6(e) with increased aerosol number concentrations, indicating that the horizontal transport of DMS due to its longer lifetime in an atmosphere with PI-oxidants is important for the increase in aerosol number concentration. Higher up in the atmosphere (above $\sim$850 hPa), the formation rate of aerosols also increases over the emission sources and at higher latitudes (not shown). The change in the total vertically integrated coagulation sink decreases by 17.7 % when switching from PD- to PI-oxidants in the PI-simulation, favoring enhanced formation of new aerosols. As the lifetime of the precursor gases and the cloud amount increases, the total deposition rate of $SO_2$ increases with 7.4 % (DMS, isoprene and monoterpene are only lost through atmospheric chemistry), favoring a decrease in the formation of new aerosols. As a result of all the competing effects, the total vertically integrated formation of new aerosols increases by 5.4 %.

Some of the newly formed $SO_4$ and SOA are lost through coagulation with the background aerosols. This coagulation sink is also reduced (-3.6 %) when switching from PD- to PI-oxidants for the same reasons as for the particles between 2 and 23.6 nm, contributing to the change in the aerosol number concentration seen in Fig. 4(a).

Even though Fig. 6 shows that the increased lifetime of the precursor gases partly can explain why the aerosol number concentration increases when switching from PD- to PI-oxidants, other factors could also play a role. The precursor gases have the potential of being oxidized in three different ways, resulting in different amounts of the end products $H_2SO_4$, $SOA_{LV}$ and $SOA_{SV}$. While both $H_2SO_4$ and $SOA_{LV}$ can take part in nucleation (to nucleation mode $SO_4$ and nucleation mode SOA, respectively), $SOA_{SV}$ can only condense onto already existing particles. If changes in the oxidation pathways favor more production of $H_2SO_4$ or $SOA_{LV}$, it can contribute to the increase in the aerosol number concentration seen in Fig. 4(a). The left panels of Fig. 7 show the contribution of the different reactions to the oxidation of the precursor gases. The largest change in the oxidant level when switching from PD- to PI-oxidants is found for $NO_3$ in the northern hemisphere (Fig. 1(c)). When switching to PI-oxidants, the relative fraction of DMS, isoprene and monoterpene oxidized by $NO_3$ is reduced (Fig. 7(a,c,d), red curves), while the oxidation involving the other oxidants become more important. For DMS, Fig. 7(a) shows that this change in the oxidation pathway will reduce the formation of species that can take part in nucleation since some of it will be converted to $SOA_{SV}$ instead of $SO_2$ (that later becomes $H_2SO_4$). For monoterpene, switching to PI-oxidants favors an oxidation pathway that gives more $SOA_{LV}$ (Fig. 7(d)), thus favoring an increase in the aerosol number concentration. An overview of all the conversion rates for the oxidation reactions in the two simulations with different oxidants is found in Table 5. Even though the global burden of nucleation mode $SO_4$-aerosols increases (+0.00650 Tg, +8.8 %), Table 5 shows that the production of $H_2SO_4$ decreases when switching from PD- to PI-oxidants (-0.5 Tg yr$^{-1}$), indicating that a shift towards more production of $H_2SO_4$ that can nucleate is not an explanation for the increase in the aerosol number concentration seen in Figure 4(a). The global burden of nucleation mode SOA-aerosols is also increasing (+0.00450 Tg, +12 %). Contrary to the case of $SO_4$, Table 5 shows that this could partly be due to a shift towards more production of a gas that can take part in nucleation since the production of $SOA_{LV}$ increases (+1.63 Tg yr$^{-1}$). Sensitivity tests in Sect. 4.3 will show that this increase in production of $SOA_{LV}$ has a negligible impact on the results in this study.

### 4.1.2   The increase in cloud droplet number concentration

Figure 4(b) shows that the CDNC increases in regions that experience large relative changes in the aerosol number concentration (Fig. 4(a)). The aerosol number concentration and CDNC increases are linked to the extended DMS lifetime discussed above (Fig. 6(b)), which in turn allows for more DMS transport to and subsequently increased aerosol formation in remote regions like the South Pacific (SP) and the Arctic Ocean (AO), as defined in Fig. 8. The region named North Pacific (NP) in Fig. 8 experiences a local minimum in the change in the aerosol number concentration. Figure 6 shows that this is caused by less aerosol formation in this region. Nevertheless, NP also experiences a relatively large increase in CDNC. The vertical profiles in Fig. 9 show that the regions which receive more precursor gases with PI-oxidants (AO and SP) experience an increase in both aerosol number concentration and CDNC for all altitudes, while the NP region experiences a decrease close to the surface, but an increase higher aloft. The latter can be explained by the vertical shift in the oxidation (Fig. 5). In NP, the height above

which the change in CDNC is positive is located lower down in the atmosphere than the height at which the aerosol number concentration starts to increase (Figs. 9(i) and 9(l)). This can be explained by the change in the size of the aerosols (Fig. 9(j)), caused by the increased aerosol condensate relative to the aerosol number concentration (Fig. 9(k)). The relative amount of condensate increases in the global mean (Fig. 9(c)) and in the northern hemisphere (Fig. 9(g) and 9(k)) because of the strong shift in the importance of the different oxidation reactions (Fig. 7). This means for DMS, the dominant precursor gas over the remote oceans, that instead of mostly getting $1 \cdot SO_2$ and no SOA from an oxidation of DMS through (R4), the PI-atmosphere will to a larger extent produce $0.75 \cdot SO_2$ and some SOA through (R3). After $SO_2$ has been oxidized to $H_2SO_4$, it nucleates easier than SOA, and 80 % of the SOA from (R3) comes as $SOA_{SV}$, which is only allowed to condense. The change in aerosol size in SP (Fig. 9(n)) deviates from the other regions. This is due to the increase in OH in SP when switching to PI-oxidants (blue colors in Fig. 1(a)), giving rise to enhanced nucleation of small $SO_4$-aerosols followed by an enhanced $H_2SO_4$-production through (R1). This also happens in AO, where the OH-level also is larger in PI, but here this effect is small relative to the effect of the increased $SOA_{SV}$-production due to the large $NO_3$-change in the northern hemisphere (Fig. 1(c)).

### 4.1.3 The change in aerosol indirect effect

The SW radiative effect of a change in CDNC varies depending on where these changes take place. Twomey (1991) showed that dA/d(CDNC), where A is the cloud albedo, is largest in clean regions with low CDNC and where the cloud albedo is approximately 0.5. The SW radiative effect will also be larger in areas with low surface albedo, in areas close to the equator due to more incoming solar radiation, and in areas where the cloud fraction is high. The last two factors, in addition to the factors in Twomey (1991), are taken into account in Eq. (6) in Alterskjær et al. (2012) when finding a cloud-weighted suscepti-bility function. This is a hybrid between the simplified dA/d(CDNC) of Twomey and the more complex d(ERFaci)/d(CDNC), which we see in Figure 3. It only includes the first aerosol indirect effect, and not second aerosol indirect effects (such as increased lifetime, cloud amount and cloud extent). The susceptibility function gives an indication of which areas over ocean that are relatively more susceptible than others to cloud albedo changes caused by changes in CDNC. The cloud-weighted susceptibility function is normalized by its maximum value. Applying this function to three years of daily output from the PIAER_PDOXI_ORG-simulation in this study results in Fig. 10(a). Areas with high cloud-weighted susceptibility are found off the west coast of the continents and in the remote southern ocean storm tracks. The large increase in CDNC (Fig. 4(b)) in the North and South Pacific regions efficiently increases the albedo of the clouds, thus resulting in the large change in the SW indirect effect seen in Fig. 3(d). Due to less insolation in the Arctic, the cloud-weighted susceptibility in this region is low, resulting in a negligible effect on the SW indirect effect, even though this is the region that experiences the relatively largest increase in both CDNC (Fig. 4(b)), cloud fraction (Fig. 4(d)) and LWP (Fig. 4(e)) due to the oxidant changes. The LW indirect effect is not dependent on the incoming solar radiation, so the large changes in cloud properties seen in the Arctic affect the LW indirect effect. The thicker and longer-lived clouds in the simulation with PI-oxidants act to reduce the difference in LW heat-ing between the PD- and PI-simulations (Fig. 3(e)). Figure 10(b) shows the vertical profile of the global mean cloud-weighted susceptibility. It shows that the decrease in CDNC close to the surface (Fig. 9(d)) does not affect the cloud albedo as much as the increase in CDNC between 900 and 800 hPa.

## 4.2 Decomposing the oxidant change

To get a better understanding of the results in the original experiment, results from the sensitivity tests where only one oxidant at a time was changed are analyzed. Figure 11 shows differences in the global mean shortwave and longwave indirect effect between the setups with modified PI-simulations (PIOXI, PIOH, PIO3, PINO3 and PIHO2) and the original setup with only PD-oxidants in both simulations. Figure 12 shows the same for the horizontal distribution. Changing only $NO_3$ (PINO3) gives almost the same result as changing all of the oxidants (PIOXI), indicating that the historical change in $NO_3$ is the most important oxidant change for indirect effect calculations. This corresponds well with Fig. 1 showing that $NO_3$ is the oxidant that has experienced the largest relative change since PI, and Fig. 7 showing that the importance of the oxidation reactions involving $NO_3$ drops the most when switching from PD- to PI-oxidants in the PI-simulation. The negative pattern over land in the tropics in PINO3 that is missing in PIOXI (Fig. 12) seems to be explained by the changes in $O_3$. Analysis of the PIO3-simulation shows that replacing only the $O_3$-oxidant with PI-values reduces the importance of (R6) where monoterpene is oxidized by $O_3$ giving $SOA_{LV}$, while the other oxidation reactions of monoterpene giving $SOA_{SV}$ become more important. This results in less new aerosol formation and increased growth of the already existing aerosols through condensation, increasing the CCN-concentration and the following cloud droplet activation and CDNC.

Table 6 shows that there are some non-linearities associated with changing one oxidant at a time. The odd numbered rows show the impact on the indirect effects when changing one oxidant at a time, while the even rows show the difference in the effect of changing all oxidant and changing all except for one oxidant. If there were no non-linearities involved in the oxidant chemistry, an odd numbered row and the following row would have shown the same numbers. This is not the case, but the differences are relatively small, supporting the indication that the contributions to the total result mainly stem from the historical changes in $NO_3$.

## 4.3 Sensitivity tests

Due to nonlinear processes and feedbacks in the model, it is difficult to separate the different effects and to estimate how much each of them contributes to the final result. As an example, enhanced formation of new aerosols can be explained as in Sect. 4.1.1, starting by the increase in lifetime of the precursor gases, but the enhanced importance of reactions giving SOA with sufficiently low volatility to nucleate new aerosols ((R3) and (R6)) can also be a part of the explanation. To get a better understanding of the importance of the various factors and processes, extra sensitivity tests with six new setups were carried out. All the test consists of three different simulations, as illustrated in Fig. 2. They all deviate from the original setup as well as from Kirkevåg et al. (2018), either through changes in some of the chemical reactions (R1-R10), manipulating the aerosol input to the code for cloud droplet activation, manipulating the code that treats the oxidants or changing the constraint on the meteorology. Information about the setups for the sensitivity tests is found in Table 7.

### 4.3.1 NOSOALVDMS and NOSOALVBVOC

When moving from a high $NO_3$-regime (PD-oxidants) to a low $NO_3$-regime (PI-oxidants), the oxidation reactions giving $SOA_{LV}$ as a product ((R3) and (R6)) become more important. This is seen from the large change in the global mean column burden of $SOA_{LV}$ (+49.6 %). Since $SOA_{LV}$ can take part in nucleation and can give rise to the increased aerosol number concentration seen in Fig. 4(a), the additional $SOA_{LV}$ that is produced when using PI-oxidants may explain the change in the indirect effects seen in Fig. 3. When replacing all of the standard produced $SOA_{LV}$ from the DMS-oxidation in (R3) with $SOA_{SV}$ in the NOSOALVDMS-simulations, the change in the total aerosol indirect effect is almost the same as for the original setup ($\Delta AIE_{tot}$: +0.25 $Wm^{-2}$), and the geographical pattern looks largely the same (not shown here). This also holds when doing the same for the oxidation of monoterpene (R6) ($\Delta AIE_{tot}$: +0.26 $Wm^{-2}$). The pattern of the resulting AIE from the oxidant changes in the NOSOALVBVOC-simulations looks almost the same as for the original simulations, except over the Amazon where the signal from the $O_3$-changes explained in the last section is gone. This does not change the global mean AIE by more than 0.01 $Wm^{-2}$, however. These sensitivity tests indicate that even though the global mean burden of $SOA_{LV}$ changes a lot when using PI-oxidants, this plays a minor role for the change in the indirect effects seen in Fig. 3.

### 4.3.2 NOSOA

The increased production of total SOA(g) ($SOA_{SV}$ and $SOA_{LV}$) when switching from PD to PI-oxidants has the potential to cause changes in the indirect effects even though the nucleation effect is negligible. All SOA(g) can condense onto already nucleated aerosols and make it easier for them to grow to the critical size for cloud droplet activation, except for cases where the reduction in hygroscopicity is more important than the increase in size. The impact of the hygroscopicity changes due to the changes in the oxidant levels has been tested and found to be negligible (not shown here). The change in total global mean column burden of SOA(g) due to changes in the oxidant level with the original setup was +40.7 %. To find out whether this increase is causing the change in the indirect effects seen in Fig. 3, the model was run with the NOSOA-setup described in Table 7. This resulted in a change in the total aerosol indirect effects ($\Delta AIE_{tot}$) of +0.14 $Wm^{-2}$, deviating by more than 0.10 $Wm^{-2}$ from the original setup. Removing products from the reaction makes the atmosphere cleaner, thus creating a different regime both for aerosol growth through reduced competition for condensable gases as for aerosol activation through reduced competition for water vapour. This means that one cannot conclude that 0.11 $Wm^{-2}$ of the 0.25 $Wm^{-2}$ is caused by an increase in condensable SOA(g) when switching from PD- to PI-oxidants, but this sensitivity test indicates that it may have contributed to the overall result seen in Fig. 3.

### 4.3.3 NACTOFF

This test is performed in order to see how important the change in the droplet activation on the smallest aerosols is. When modifying the oxidant level, the smallest aerosols are affected by the change in formation rate, while all aerosols are affected by the change in condensation. The results from this test give an indication of how important the changes associated with the smallest aerosols are. When not allowing the smallest aerosols in mixture number 1 (corresponding to the nucleation

mode in modal aerosol schemes) to activate, the change in the total aerosol indirect effects found when switching from PD- og PI-oxidants in the PI-simulation is small ($\Delta AIE_{tot}$: -0.03 Wm$^{-2}$). This confirms that it is the difference in the number concentration of the smallest SO$_4$- and SOA-aerosols between the simulations with different oxidant levels that gives the large difference in the indirect effect seen in Fig. 3.

### 4.3.4  DIURNALNO3

The tests where the oxidant changes where studied individually identified the historical change in NO$_3$ as having the largest impact on the result. As described in the model description, OH and HO$_2$ have a diurnal cycle added to the prescribed monthly climatology in CAM5.3-Oslo. This is not the case for NO$_3$, even though it is well known that concentrations of NO$_3$ drop during daytime due to rapid photolysis (Wayne et al., 1991; Seinfeld and Pandis, 2016). To see how this lack of a diurnal cycle for NO$_3$ impacts the results in this study, another set of simulations was carried out. The daytime concentration of NO$_3$ was set to zero, while the nighttime concentration was increased, such that the daily averaged and the monthly averaged values stayed the same as in the original setup. This treatment of the diurnal cycle is the same as that for HO$_2$ and OH, but with a shift from day to night. Carrying out the same three model simulations with this new setup as for the original default model setup gives a change in the total aerosol indirect effect of +0.26 Wm$^{-2}$ (from -1.32 Wm$^{-2}$ to -1.06 Wm$^{-2}$) when applying PI- instead of PD-oxidants. In other words, this test shows that the lack of a diurnal cycle for NO$_3$ only has a minor influence on the result in this study. The reason for this minor impact is that the main effect of oxidation by NO$_3$ is of DMS over the oceans. Since the lifetime of DMS is 36 and 55 hours (present-day and preindustrial respectively), the reduction in the nighttime oxidation when not applying a diurnal cycle will have time to be compensated by an increase in the daytime oxidation.

### 4.3.5  FREEMET

Constraining the natural variability by nudging the meteorology has been shown to be an efficient way of identifying the effect of a model perturbation since it reduces the computational cost and time significantly (Kooperman et al., 2012). In this study, nudging has been applied in order to model ERF$_{aci}$. According to the definition of effective radiative forcing in Myhre et al. (2013, p. 665), "ERF represents the change in net TOA downward radiative flux after allowing for atmospheric temperatures, water vapor and clouds to adjust, but with global mean surface temperature or a portion of surface conditions unchanged". In the simulations presented here, nudged winds are not fully impacted by rapid adjustments in the atmosphere due to an aerosol perturbation, which again could give a response by the clouds. Thus, parts of this rapid wind-aerosol-cloud-radiation feedback could be missing from the calculated values of ERF in this study. Running all the simulations in this study with free meteorology is computationally very expensive. Instead we performed sensitivity tests for the three simulations with the original model setup to estimate the bias introduced by the method of nudging. The length of the simulations is 53 years, where the last 50 are analyzed. The total aerosol indirect effect changes by 0.3 ± 0.2 Wm$^{-2}$ (from -1.3 ± 0.2 Wm$^{-2}$ to -1.0 ± 0.2 Wm$^{-2}$) when switching from PD- to PI-oxidants in the PI-simulation. Even though the uncertainties due to natural variability still are large after 50 years, this change in the total aerosol indirect effect due to historical oxidant changes fall in the same

range as when nudging the winds with a relaxation time scale of six hours. Analyzing only the last 30 years of the simulations gives the same change in the total aerosol indirect effect, indicating that there is no drift in the signal.

## 5 Summary and conclusions

We have used the global atmospheric model CAM5.3-Oslo to study the effect of historical oxidant changes on the PD-PI aerosol indirect effect. The precursor gases in the PI-simulation were exposed to PI-oxidants instead of PD-oxidants. Our main findings are:

- The total aerosol indirect effect is reduced from -1.32 Wm$^{-2}$ to -1.07 Wm$^{-2}$, mainly due to a cloud brightening in the modified PI-simulation.

- NO$_3$ is the oxidant that contributes the most to the changes.

- When the precursor gases are exposed to an atmosphere with relatively lower oxidative power (PI-oxidants vs. PD-oxidants), their lifetimes increase and they are transported higher up in the atmosphere and horizontally towards more remote areas before they are oxidized.

- The increased lifetime of the precursor gases contributes to an increase in the formation of new aerosol and a decrease in the deposition and in the coagulation sink of the newly formed aerosols, contributing to an increase in the aerosol number concentration.

- A large portion of the new aerosol formation and the increase in aerosol number concentration occurs where the cloud-weighted susceptibility is high, giving a large impact on the radiative effects.

- The change from PD- to PI-oxidants in the PI-simulation yields a shift in the chemical reactions towards increased production of condensate relative to the amount of gases that can nucleate, which increases the size of the aerosols, making it easier for them to activate.

Note, that the magnitude of the sensitivity of the total aerosol indirect effect to the choice of the oxidants in this study is as large as the total sulfur direct forcing (Myhre et al., 2013), thus contributing significantly to the total preindustrial-to-present-day anthropogenic forcing. Overviews of model results of the PD-PI aerosol indirect effect show occasionally so negative values that they even offset the warming from the greenhouse gases (Boucher et al., 2013; Lohmann, 2017). Our results suggest that such unrealistic cooling may appear less often if the precursor gases are exposed to oxidants of their era, instead of applying PD-oxidants for both PD- and PI-simulations.

The results in this study are based on simulations from just one model, with its model-specific treatments of oxidants, aerosols, clouds and radiation that all include uncertainties and simplifications. This also holds for the single input dataset used for the prescribed oxidants. An evaluation of the extent to which uncertainties in the different parameterizations and in the prescribed oxidant fields affect the result is beyond the scope of this paper, but should be focus for future studies. The treatment

of the MSA-product from DMS-oxidation by OH (R3) should be looked at in particular, since the changes in SOA-condensate from that reaction seem to contribute to the resulting changes in the total aerosol indirect effect. Different choices of yields for the oxidation reactions in Table 2 should also be in focus since these yields are uncertain and vary between different models and observations (Kroll et al., 2005; Lee et al., 2006; Dentener et al., 2006; Spracklen et al., 2011; Neale et al., 2012; Tsigaridis et al., 2014; Jokinen et al., 2015). The impact of the lack of pure biogenic new particle formation in the model applied in this study could also be studied, since this mechanism has been shown to be important for radiative forcing calculations, especially in clean regions (Gordon et al., 2016). When it comes to the oxidant input dataset, it would be interesting to see how the result is affected by using a model with online oxidant chemistry. Upcoming studies should also see how the result is affected by using other input datasets produced by more advanced models than the model applied in Lamarque et al. (2010), which for example did not include online aerosol-cloud radiative interactions or different land cover information in the two different eras, which could have impacted the oxidant level through different photolysis rates and different emissions from the land model.

The impact of the oxidant changes also depends on the emissions of precursor gases. Carslaw et al. (2013) show that there are large uncertainties linked to natural emissions, even when assuming that they do not vary between PI and PD. This was shown especially for DMS (Woodhouse et al., 2010), which is found to be one of the most important precursor gases in this study. Changes in temperature and pH in the ocean, as well as changes in land use, insolation and $CO_2$ may also have contributed to a change in the emissions since preindustrial time (Charlsson et al., 1987; Guenther et al., 2012; Unger, 2014). CAM5.3-Oslo is also lacking some emissions that could be important for the magnitude of the effect of the oxidant changes, for example emissions of BVOC from the ocean, which can contribute significantly to the marine aerosol loading (Shaw et al., 2010), creating a more polluted regime with the potential of different susceptibilities.

Despite the large uncertainties and simplifications mentioned above, we find that the treatment of the oxidants is open for discussion. We suggest that a common way of treating the oxidants must be agreed upon when modeling aerosol effective radiative forcings. We also encourage other researchers to specify which oxidants are used in their studies of historical changes in aerosol-cloud interactions.

Simulations from the Aerosol Chemistry Model Intercomparison Project (AerChemMIP), endorsed by the Coupled-Model Intercomparison Project 6 (CMIP6) can be used to quantify preindustrial-to-present-day effective radiative forcings. Comparing the cloud forcings from the simulations called piClim-aer and piClim-control (Collins et al., 2017) will be approximately the same as done in the original default setup in this study, with the same oxidant level in both simulations. For models without tropospheric chemistry, AerChemMIP does not include a setup that takes into account historical oxidant changes. However, models that include tropospheric chemistry can perform the simulation piClim-NTCF, which includes different ozone precursors in the two different simulations, giving a different oxidation capacity. The piClim-NTCF simulation does not include all the factors that contribute to the differences in the oxidant level between PD and PI (for example methane), but it includes some of them, so we suggest that a comparison of the cloud forcings in piClim-NTCF and piClim-control will facilitate calculations of the PD-PI aerosol indirect effect, including changes due to different oxidant level, also for the CMIP6-models.

*Competing interests.* The authors declare that they have no conflict of interest.

*Acknowledgements.* I. H. H. K., A. G., D. O., A. K., Ø. S., T. I. and M. S. have been financed by the research council of Norway (RCN) through the project EVA and the NOTUR/Norstore projects (Sigma2 account: nn2345k, Norstore account: NS2345K). We gratefully acknowledge Sara Marie Blichner and Moa Sporre for scientific discussions.

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

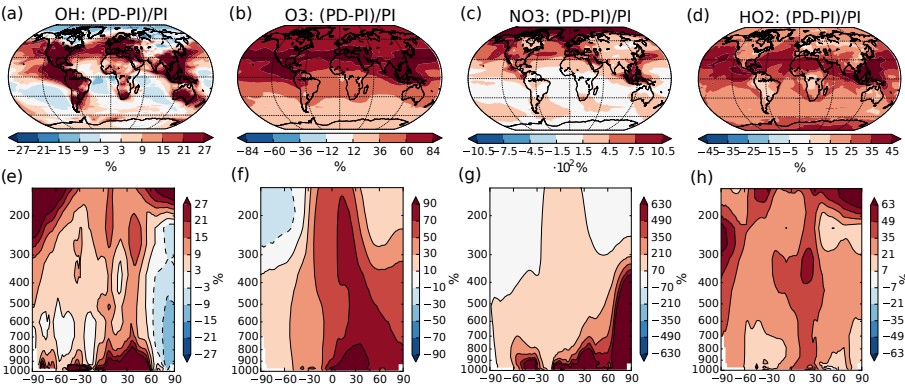

**Figure 1.** Percent-wise change in the annual mean oxidant mixing ratio (mol/mol) between PI and PD in the dataset from Lamarque et al. (2010) used in this study. Top: mean change from surface and up to 550 hPa. Bottom: zonal mean change. Please note the different scales on the color bars.

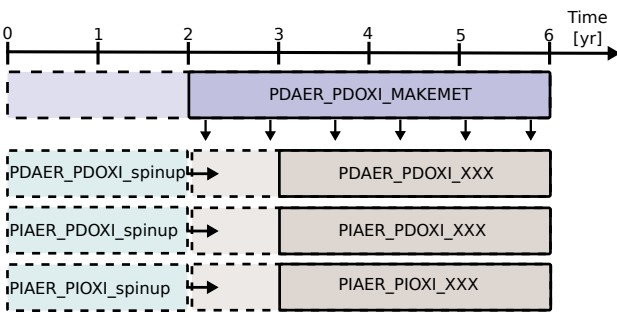

**Figure 2.** Overview of how the simulations were carried out. PDAER_PDOXI_MAKEMET produced meteorology for the other simulations from its last four years. Dashed lines show the part of the simulations used as spin-up. Horizontal arrows show that the simulations to the right of the arrow restarted from the already spun up simulation to the left. The spin-up cases were not nudged, but started with free running meteorology from the same state as PDAER_PDOXI_MAKEMET. XXX refers to either ORG (original model setup), or the name of the sensitivity tests described in Sect. 3.2 and 4.3.

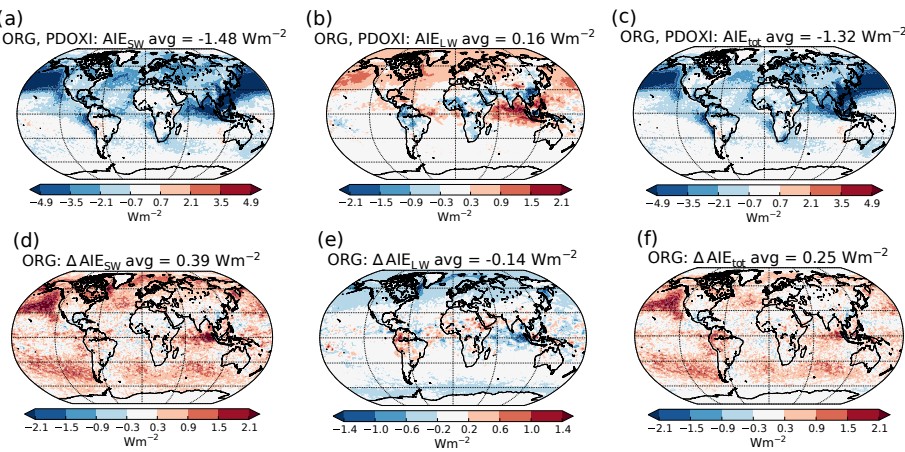

**Figure 3.** Top: PD-PI aerosol indirect effect when using the standard setup with PD-oxidants in both simulations. Left: shortwave, middle: longwave, right: total. Bottom: differences in the PD-PI indirect effect between simulations performed with PI- and PD-oxidants in the PI-simulation.

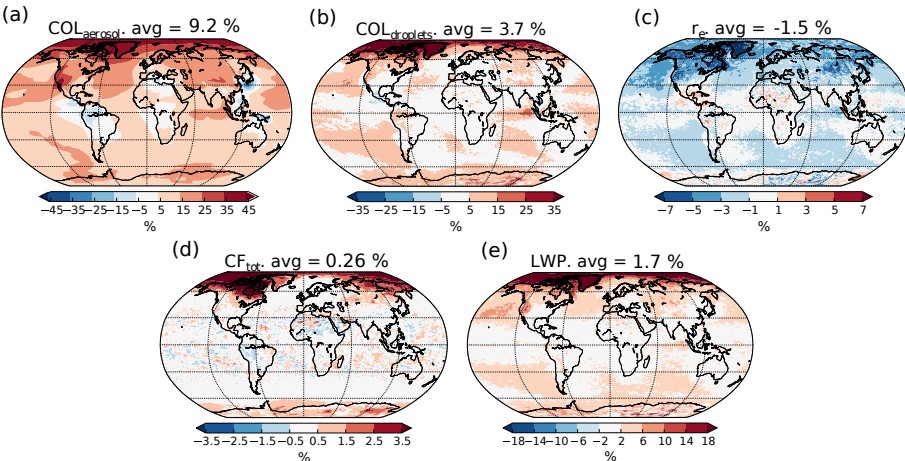

**Figure 4.** Relative change in aerosol and cloud properties in the PI-simulation when switching from PD- to PI-oxidants. (a) Column number of aerosols, (b) column number of cloud droplets, (c) effective radius of cloud droplets in the cloud top layer, (d) total cloud fraction, and (e) total gridbox averaged liquid water path.

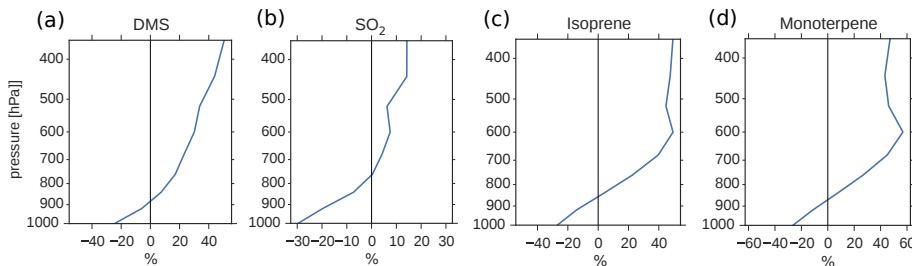

**Figure 5.** Global mean relative change in chemical loss of (a) DMS, (b) SO$_2$, (c) isoprene and (d) monoterpene when switching from PD- to PI-oxidants in the PI-simulation.

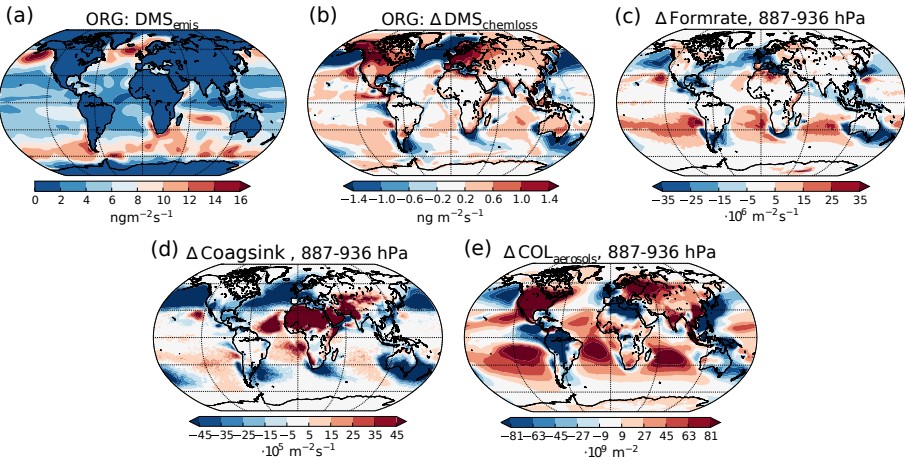

**Figure 6.** (a) Emission rate of DMS (same for both PI-simulations). (b) Difference in net chemical loss of DMS though oxidation (c) Difference in aerosol formation rate in the layer 887-936 hPa. (d) Difference in the coagulation sink during nucleation in the layer 887-936 hPa. (e) Difference in column burden of aerosols in the layer 887-936 hPa. All differences show values from the PI-simulation using PI-oxidants minus values from the PI-simulation using PD-oxidants.

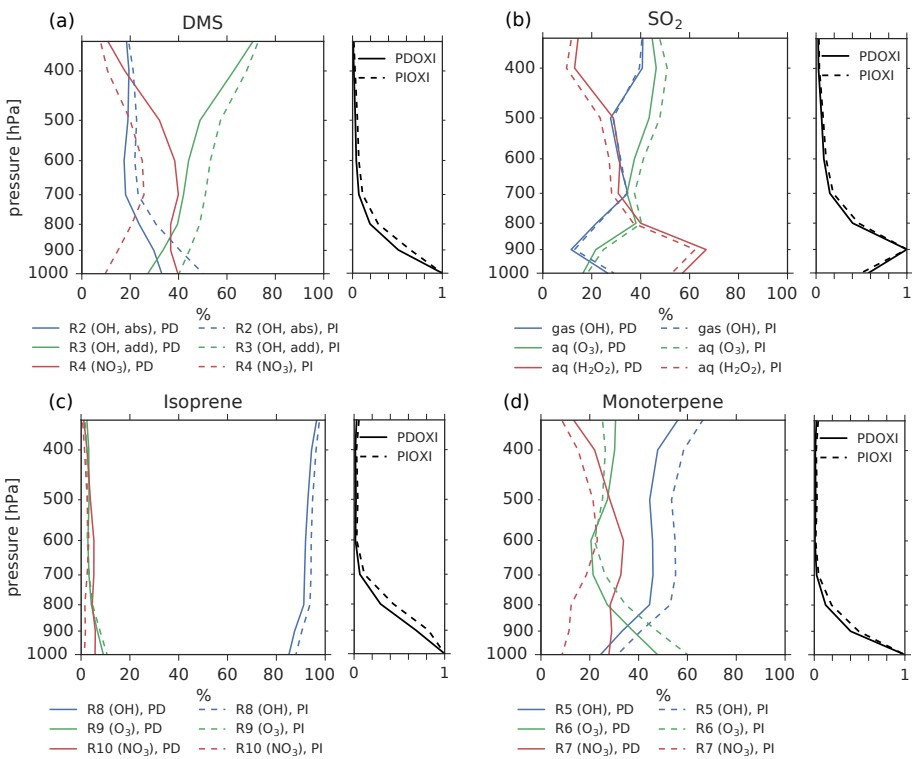

**Figure 7.** The left panel of each figure shows the importance of different oxidant reactions at different levels for (a) DMS, (b) SO$_2$, (c) isoprene and (d) monoterpene. Solid lines: PD-oxidants, dashed lines: PI-oxidants. The curves indicate the percentage of the total oxidation for each specie that occurs through the specified reactions at a specific height. The sum of the three reactions at each level is equal to 100 % in all cases. The right panel of each figure shows how much of the specie is oxidized at each level relative to the level of maximum oxidation.

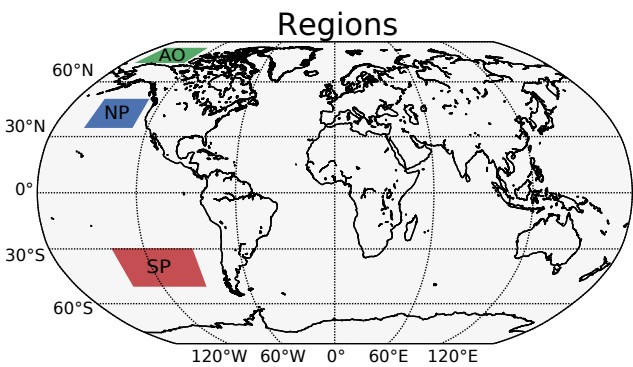

**Figure 8.** Selected regions with extra focus. AO: Arctic Ocean (70° N - 82° N, 130° W - 170° W). NP: North Pacific (35° N - 50° N, 130° W - 160° W). SP: South Pacific (30° S - 50° S, 90° W - 140° W).

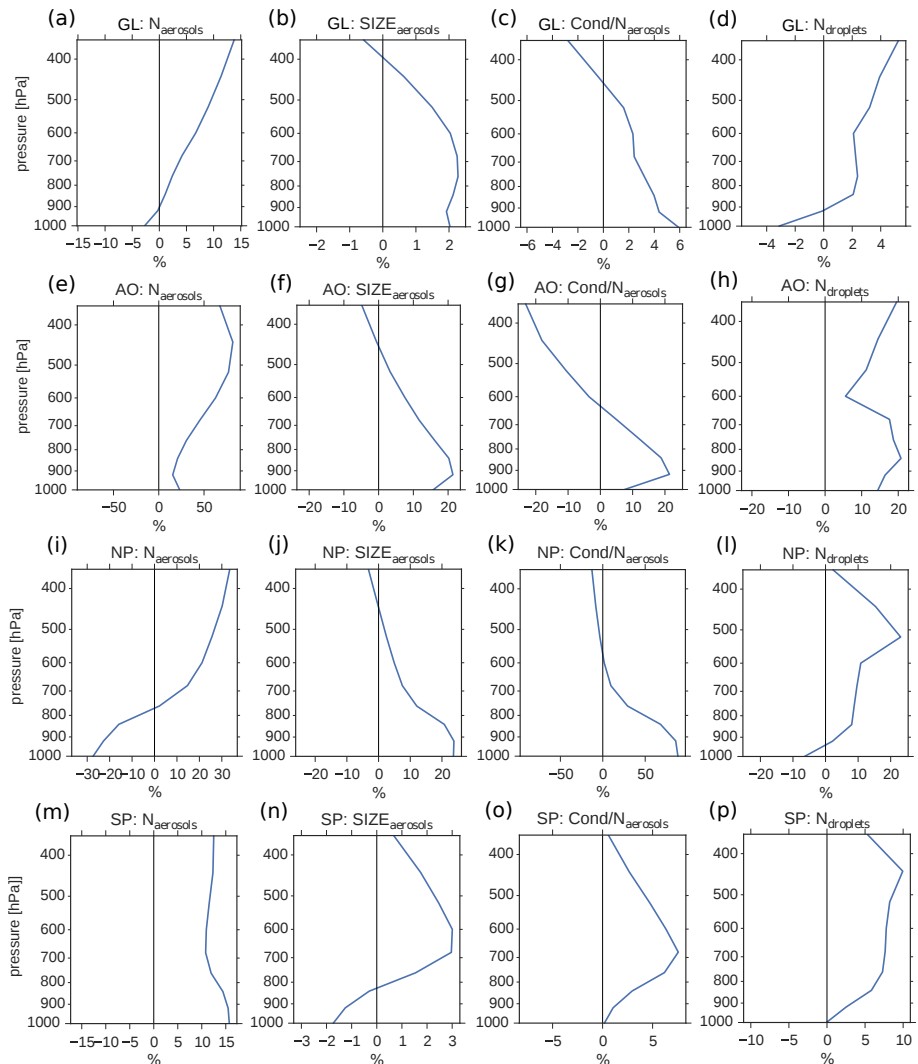

**Figure 9.** Vertical profiles of annual averaged changes in aerosol number concentration (left), aerosol size (middle left), aerosol condensate divided by the aerosol number concentration (middle right) and CDNC (right) on a global mean (GL) and in the three different regions from Fig. 8 (Arctic Ocean (AO), North Pacific (NP) and South Pacific (SP)), when switching from PD- to PI-oxidants in the PI-simulation. The mean size of the aerosols in the middle left column is calculated as a mean of the number mean radius of all mixtures in the model, weighted by the number of aerosols in each mixture.

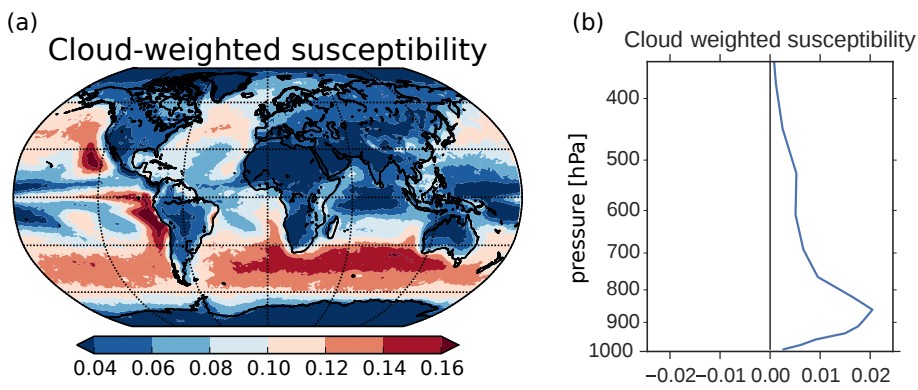

**Figure 10.** (a) Cloud-weighted susceptibility using Eq. (6) in Alterskjær et al. (2012). Cloud droplet size and numbers from the cloud top layer and the total cloud fraction were applied. (b) Vertical profile of the global mean cloud-weighted susceptibility.

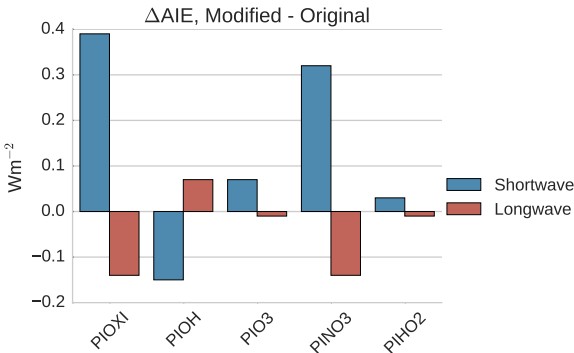

**Figure 11.** Differences in global mean shortwave and longwave aerosol indirect effect between the setups with modified PI-simulations (PIOXI, PIOH, PINO3 and PIHO2) and the original setup.

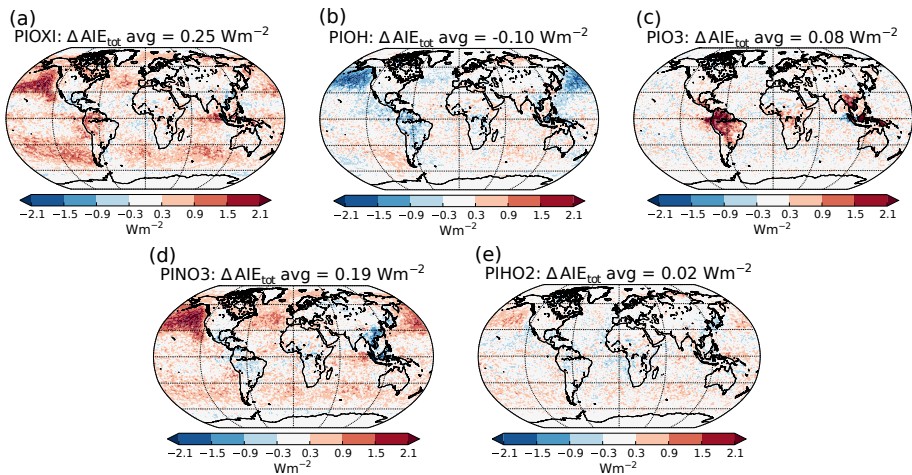

**Figure 12.** Differences in total aerosol indirect effect between the PI-simulation with (a) PIOXI, (b) PIOH, (c) PIO3, (d) PINO3, (e) PIHO2 and the original PI-simulation with only PD-oxidants.

**Table 1.** Overview of global emission rates and burdens of the precursor gases in CAM5.3-Oslo. The values come from three different simulations using aerosols and oxidants from present-day, (aerosols from preindustrial and oxidants from present-day), and {aerosols and oxidants from preindustrial}.

| Specie | Emission rates [Tg yr$^{-1}$] | Burdens [Tg] |
|---|---|---|
| SO$_2$ | 130 | 0.705 |
| | (29.0) | (0.319) |
| | {29.0} | {0.380} |
| DMS | 66.3 | 0.276 |
| | (66.2) | (0.274) |
| | {66.2} | {0.417} |
| Isoprene | 406 | 0.148 |
| | (418) | (0.150) |
| | {417} | {0.287} |
| Monoterpene | 114 | 0.0358 |
| | (116) | (0.0341) |
| | {116} | {0.0697} |

**Table 2.** Chemical reactions with corresponding rate coefficients. For (R1), $fc = 3 \cdot 10^{-31} \cdot \left(\frac{300}{T}\right)^{3.3}$, and $ko = \frac{fc \cdot M}{1 + (fc \cdot M \cdot 1.5 \cdot 10^{12})}$, where $M$ is the number concentration of all molecules that can act as a third body [cm$^{-3}$]. If the model does not trace an end product of a chemical reaction, the product is lost in the model and not written down in this table, explaining why the stoichiometry is not exact in all of the reactions.

| Reaction number | Reaction | Rate coefficient [cm$^3$molecule$^{-1}$s$^{-1}$] |
|---|---|---|
| (R1) | $SO_2 + OH + M \rightarrow H_2SO_4 + M$ | $ko \cdot 0.6^{\left(1 + \left(\log_{10}(fc \cdot M \cdot 1.5 \cdot 10^{12})\right)^2\right)^{-1}}$ |
| (R2) | $DMS + OH \rightarrow SO_2$ | $9.6 \cdot 10^{-12} \cdot e^{-234/T}$ |
| (R3) | $DMS + OH \rightarrow 0.75 \cdot SO_2 + 0.5 \cdot HO_2 + 0.029 \cdot SOA_{LV} + 0.114 \cdot SOA_{SV}$ | $\frac{\left(1.7 \cdot 10^{-42} \cdot e^{7810/T}[O_2]\right)}{\left(1 + 5.5 \cdot 10^{-31} e^{7460/T}[O_2]\right)}$ |
| (R4) | $DMS + NO_3 \rightarrow SO_2 + HNO_3$ | $1.9 \cdot 10^{-13} \cdot e^{-520/T}$ |
| (R5) | $monoterpene + OH \rightarrow 0.15 \cdot SOA_{SV}$ | $1.2 \cdot 10^{-11} \cdot e^{-440/T}$ |
| (R6) | $monoterpene + O_3 \rightarrow 0.15 \cdot SOA_{LV}$ | $8.05 \cdot 10^{-16} \cdot e^{-640/T}$ |
| (R7) | $monoterpene + NO_3 \rightarrow 0.15 \cdot SOA_{SV}$ | $1.2 \cdot 10^{-12} \cdot e^{-490/T}$ |
| (R8) | $isoprene + OH \rightarrow 0.05 \cdot SOA_{SV}$ | $2.7 \cdot 10^{-11} \cdot e^{-390/T}$ |
| (R9) | $isoprene + O_3 \rightarrow 0.05 \cdot SOA_{SV}$ | $1.03 \cdot 10^{-14} \cdot e^{-1995/T}$ |
| (R10) | $isoprene + NO_3 \rightarrow 0.05 \cdot SOA_{SV}$ | $3.15 \cdot 10^{-12} \cdot e^{-450/T}$ |
| (R11) | $HO_2 + HO_2 \rightarrow H_2O_2$ | $\left(3.5 \cdot 10^{-13} \cdot e^{430/T} + 1.7 \cdot 10^{-33} \cdot e^{1000/T}\right)$ $\cdot \left(1 + 1.4 \cdot 10^{-21} \cdot [H_2O] \cdot e^{2200/T}\right)$ |
| (R12) | $H_2O_2 + OH \rightarrow H_2O_2 + HO_2$ | $2.9 \cdot 10^{-12} \cdot e^{-160/T}$ |
| (R13) | $H_2O_2 + h\nu \rightarrow 2 \cdot OH$ | |

**Table 3.** Overview of the prescribed precursor- and aerosol emissions and prescribed oxidant concentrations used in the three different simulations that were carried out for each modification to the default model setup.

| Name of simulations | Prescribed emissions of aerosols and precursor gases | Prescribed concentrations of oxidants | SSTs, sea-ice extent, greenhouse gases and land use |
|---|---|---|---|
| PDAER_PDOXI_XXX | PD | PD | PD |
| PIAER_PDOXI_XXX | PI | PD | PD |
| PIAER_PIOXI_XXX | PI | PI | PD |

**Table 4.** Global mean lifetime of different gaseous and aerosol species (g: gas, a: aerosol) when applying PD- and PI-oxidants in the PI-simulation. The lifetime is calculated as (Global mean burden)/(Global mean loss).

| Species | Lifetime, PD [h] | Lifetime, PI [h] | Change in lifetime [%] |
|---|---|---|---|
| $SO_2$ (g) | 29 | 34 | +17 |
| DMS (g) | 36 | 55 | +53 |
| Isoprene (g) | 3.2 | 6.0 | +88 |
| Monoterpene (g) | 2.6 | 5.3 | +104 |
| $H_2SO_4$ (g) | 0.91 | 1.0 | +9.9 |
| $SOA_{LV}$ (g) | 0.65 | 0.82 | +26 |
| $SOA_{SV}$ (g) | 0.75 | 1.0 | +9.9 |
| $SO_4$ (a) | 78 | 84 | +7.7 |
| SOA (a) | 115 | 116 | +0.9 |

**Table 5.** Conversion rates using present-day (preindustrial) oxidants.

| Reaction | | | Loss [Tg yr$^{-1}$] | | Production [Tg yr$^{-1}$] | |
|---|---|---|---|---|---|---|
| (R2) | DMS + OH | [DMS] | 24.0 | $\rightarrow$ | 24.7 | [SO$_2$] |
| | | | (31.4) | | (32.4) | |
| (R3) | DMS + OH | [DMS] | 20.6 | $\xrightarrow{0.75}$ | 16.0 | [SO$_2$] |
| | | | (26.7) | | (20.7) | |
| | | | | $\xrightarrow{0.029}$ | 1.62 | [SOA$_{LV}$] |
| | | | | | (2.10) | |
| | | | | $\xrightarrow{0.114}$ | 6.38 | [SOA$_{SV}$] |
| | | | | | (8.26) | |
| (R4) | DMS + NO$_3$ | [DMS] | 26.3 | $\rightarrow$ | 27.1 | [SO$_2$] |
| | | | (10.4) | | (10.7) | |
| (R5) | monoterpene + OH | [monoterpene] | 41.3 | $\xrightarrow{0.15}$ | 7.65 | [SOA$_{SV}$] |
| | | | (50.6) | | (9.37) | |
| (R6) | monoterpene + O$_3$ | [monoterpene] | 45.2 | $\xrightarrow{0.15}$ | 8.38 | [SOA$_{LV}$] |
| | | | (51.4) | | (9.53) | |
| (R7) | monoterpene + NO$_3$ | [monoterpene] | 32.8 | $\xrightarrow{0.15}$ | 6.09 | [SOA$_{SV}$] |
| | | | (12.7) | | (2.36) | |
| (R8) | isoprene + OH | [isoprene] | 376 | $\xrightarrow{0.05}$ | 46.4 | [SOA$_{SV}$] |
| | | | (376) | | (46.4) | |
| (R9) | isoprene + O$_3$ | [isoprene] | 26.7 | $\xrightarrow{0.05}$ | 3.30 | [SOA$_{SV}$] |
| | | | (27.6) | | (3.41) | |
| (R10) | isoprene + NO$_3$ | [isoprene] | 21.8 | $\xrightarrow{0.05}$ | 2.70 | [SOA$_{SV}$] |
| | | | (6.72) | | (0.830) | |
| (R2) | SO$_2$ + OH + M | [SO$_2$] | 10.4 | $\rightarrow$ | 16.0 | [H$_2$SO$_4$] |
| | | | (10.1) | | (15.5) | |
| (aq) | SO$_2$ + O$_3$ | [SO$_2$] | 14.6 | $\rightarrow$ | 21.9 | [SO$_4$] |
| | | | (14.8) | | (22.3) | |
| (aq) | SO$_2$ + H$_2$O$_2$ | [SO$_2$] | 28.4 | $\rightarrow$ | 42.6 | [SO$_4$] |
| | | | (22.5) | | (33.7) | |
| | SO$_2$ dry deposition | [SO$_2$] | 16.5 | | | |
| | | | (16.5) | | | |
| | SO$_2$ wet deposition | [SO$_2$] | 22.5 | | | |
| | | | (25.4) | | | |

**Table 6.** Difference in global mean SW and LW indirect effects between setups with the modified PI-simulation in the second column and the default PI-simulation with PD-oxidants. The bottom row shows the effect of changing all of the oxidants at the same time (similar to Fig. 3(c,d)), the other odd numbered rows show the effect of changing one oxidant at the time in the PI-simulation, while the even numbered rows show the difference in switching all oxidants (PIOXI) and all but one (PIOXI_PDXXX) in the PI-simulation.

| Row number | Description of the modified PI-simulation | Change in shortwave aerosol indirect effect [Wm$^{-2}$] | Change in longwave aerosol indirect effect [Wm$^{-2}$] |
|---|---|---|---|
| 1 | PDOXI_PIOH | -0.15 | +0.07 |
| 2 | PIOXI − PIOXI_PDOH | -0.06 | +0.02 |
| 3 | PDOXI_PIO3 | +0.07 | -0.01 |
| 4 | PIOXI − PIOXI_PDO3 | +0.12 | 0.00 |
| 5 | PDOXI_PINO3 | +0.32 | -0.14 |
| 6 | PIOXI − PIOXI_PDNO3 | +0.41 | -0.11 |
| 7 | PDOXI_PIHO2 | +0.03 | -0.01 |
| 8 | PIOXI − PIOXI_PDHO2 | +0.03 | +0.01 |
| 9 | PIOXI | +0.39 | -0.14 |

**Table 7.** Information about how the setup for the sensitivity tests deviates from the default original setup. The right column shows how the total aerosol indirect effect changes when switching from PD- to PI-oxidants in the PI-simulation. $\Delta\text{AIE}_{\text{tot}}$ with the default model setup was +0.25 $\text{Wm}^{-2}$

| Name of simulations | Description of setup | $\Delta\text{AIE}_{\text{tot}}$ [$\text{Wm}^{-2}$] |
|---|---|---|
| NOSOALVDMS | None of the SOA produced through (R3) is allowed to nucleate new particles. (R3) is thus replaced with $$\text{DMS} + \text{OH} \rightarrow 0.75 \cdot \text{SO}_2 + 0.5 \cdot \text{HO}_2 + 0.143 \cdot \text{SOA}_{\text{SV}}$$ | +0.25 |
| NOSOALVBVOC | None of the SOA produced through (R6) is allowed to nucleate new particles. (R6) is thus replaced with $$\text{monoterpene} + \text{O}_3 \rightarrow 0.15 \cdot \text{SOA}_{\text{SV}}$$ | +0.26 |
| NOSOA | No SOA production from DMS-oxidation. (R3) is thus replaced with $$\text{DMS} + \text{OH} \rightarrow 0.75 \cdot \text{SO}_2 + 0.5 \cdot \text{HO}_2$$ | + 0.14 |
| NACTOFF | No activation from particle mixture number 1 (Kirkevåg et al., 2018). This mixture corresponds to the nucleation mode in modal aerosol schemes, and this is where we find the newly formed SOA- and SO$_4$-aerosols. | -0.03 |
| DIURNALNO3 | Add a daily cycle to the concentrations of NO$_3$ that come from prescribed, monthly mean values. | +0.26 |
| FREEMET | Apply free meteorology instead of nudged winds. | +0.3 $\pm$ 0.2 |