# Peer review of "Strong impacts on aerosol indirect effects from historical oxidant changes"

_Atmospheric Chemistry and Physics, 2017_

## Referee Comment (RC1) · Anonymous Referee #1 · 24 Feb 2018

This paper evaluates the impact of changing concentrations of oxidant between the preindustrial to present day on aerosol-cloud interactions. The change in oxidant, changes the oxidation pathways of emitted gas phase compounds (SO2, DMS, Isoprene, Monoterpenes) leading to change is the concentration of low/semi volatile compounds and so onto aerosol and clouds. The paper uses an offline calculation of chemistry where oxidants are specified from another model simulation. A highly simplified chemical scheme is then used to evaluate the influence of these changes on a range of aerosol and cloud properties and concludes that there is a relatively large change in aerosol indirect effects due to the change in oxidants. The paper attributes much of this change to the large increase in NO3 concentrations between the pre-industrial and the present day.

The paper is in general well written and I think the paper is potentially a useful contribution to the field. However, I have some significant issues with some of the methods used and hence the conclusions. Until these issues are resolved I feel the publication is unwarranted.

Major issues.

Diurnal Cycle of NO3. The paper identifies the increase in NO3 between the preindustrial and the present day as the major change in oxidants. This is most likely true. However, their model treats NO3 in a relatively unsophisticated manner. The text explicitly says that OH and HO2 concentrations have a diurnal cycle imposed on them whereas NO3 is not mentioned suggesting that it does not. In reality NO3 does have a significant diurnal cycle. Its rapid photolysis leads to low concentrations during the day and high during the night.

Thus, in the real atmosphere NO3 anti-correlates with isoprene and monoterpenes (this depends to an extent upon the monoterpene speciation) thus NO3 likely only plays a minor role in the oxidation of these species. By not having a diurnal cycle in the NO3 concentration this anti-correlation is lost in the model which then likely favours the NO3 oxidation route over the other oxidations routes.

Given the central importance of NO3 to the primary conclusion of the paper this lack of anti-correlation provides significant problem that needs to be resolved before the paper can go forwards to publication.

Attribution of impacts The chemistry scheme in the model is fundamental to the magnitude of the impacts simulated. There is little explanation for the choice of yield from the VOCs or DMS. A section explaining these choices and previous work to justify these choices would be useful.

There are couple of ways that the change in oxidants could impact the oxidation of the precursor compounds, and so the production of low volatile products. They could
change the net yield of compound (SOA.SV vs SOA.LV), or they could change the location of the oxidation. It would be useful to know which process is occurring here. Given that the yield of SOA.SV is the same for all isoprene oxidation routes, any change in the cloud-aerosol feedback due to isoprene can only be occurring because of the change in the location of oxidation - the globally total production of SOA.SV from isoprene is the same in all simulations (0.15 \* total isoprene emission). Monoterpenes behave differently. Oxidation by O3 leads to the production of SOA.LV whereas oxidation by other methods leads to SOA.SV. Here a change in the oxidant may lead to significant changes in the production of the different SOA times. A similar change might occur with DMS oxidation. The authors attribute all of the changes they see to changes in the location of the oxidation, but they don't really give much evidence to support this. Figure 9 shows the change in the fractional oxidation as a function of height, but this doesn't show in absolute terms how much of the precursor is oxidized in different places. This is a little misleading - very little of the monoterpene / isoprene is oxidised in the upper troposphere compared to the lower troposphere. Perhaps showing the absolute change in oxidation would help with this? This would probably need to be on a log scale.

It would be useful to have a table or figure which shows how much H2SO4, SOA.LV, and SOA.SV is produced globally by each route for each simulation.

Given the length of the description of the impact on the aerosol and clouds there should be an increase in the description of how the chemistry is making this impact. This explanation of why the emissions are changing the SOA.LV, SOA.SV and H2SO4 tracers is somewhat weak.

Minor comments.

Can the total mass of isoprene and monoterpene be included into a table somewhere? The MEGAN scheme can lead to significantly varying emissions depending upon the implementation into the transport model. It would be useful to know these values. It
would also be useful to state that the assumption all of the mono-terpene emission is considered to be alpha-pinene which seems to be the implication of the rate constants chosen.

The chemistry scheme also doesn't seem to include aqueous SO2+H2O2. Although this is occurring within the cloud phase it is still a chemical reaction and for completeness I think it should be included.

Figure 9: Figure Caption. The language in the figure caption should be re-examined as it doesn't make sense

---

## Referee Comment (RC2) · Anonymous Referee #3 · 25 Feb 2018

This excellent atmospheric modeling paper presents the revival of an important, too neglected, topic: how gas-aerosol interactions influence aerosol climate impacts. The study quantifies the sensitivity of aerosol-cloud radiative forcing (ERFaci) or "aerosol indirect effect" (preindustrial to present day) to the use of preindustrial versus present-day oxidant fields/input data. The study finds a substantial sensitivity (20-30%), a smaller ERFaci estimate when the preindustrial oxidant fields are correctly applied. The upshot is that when using PI oxidant fields, clouds are brighter in the PI era. Many published and active global modeling studies of aerosol-cloud interactions continue to apply off-line unchanging oxidant fields in the different climatic states. Over a decade ago, a line of research first showed how the anthropogenic aerosol direct radiative forcing is sensitive to changing oxidants (e.g. Berglen et al., 2004; Bell et al., 2003;

[Figure]

Unger et al., 2006). It is timely and exciting to demonstrate quantitatively the sensitivity of the ERFaci to atmospheric photochemistry. The study does a fine job to identify the chemical mechanisms responsible for this sensitivity via a set of 8 perturbation runs altering the PI or PD state of a single or set of oxidants. NO3 radical changes drive most of the sensitivity. The paper is well written and structured. The figures are all necessary, clear and properly labeled. I highly recommend publication once the following issues are addressed.

1. The preindustrial oxidant fields are from Lamarque et al., 2010. Is it correct to assume that these oxidant fields were generated without on-line aerosol-cloud radiative interactions i.e. this preindustrial gas-phase chemistry does not "see" the brighter preindustrial clouds found in the present study with CAM5.3-Oslo? This question has broader implications. The changes in aerosol-cloud interactions and associated meteorology in PI versus PD state will have an influence on the resultant oxidant levels, not least through altering photolysis rates. How does the application of off-line oxidants here versus fully 2-way coupled on-line oxidants affect the main results?

2. It is not clear how long the simulations are run for in total? However, it is reported that the last 3 years of the run are used for the analyses. ERF allows all feedbacks between land-atmosphere and the land-atmosphere system to come into steady-state with the imposed radiative perturbation. Is the land-atmosphere system in steady-state after only 3 years of running the model? Many of the global chemistry-climate model frameworks seem to run for much longer (even with fixed SSTs and sea ice) to allow for the land-atmosphere system to come into steady-state i.e. more than 20 years.

3. Is it methodologically correct to 'nudge' a simulation and calculate ERF?

4. Is it possible to use the 3 model run years to generate a standard error estimate of uncertainty based on interannual internal variability – thus not providing naked numbers e.g. -1.32 W/m2 and -1.07 W/m2. The numbers may appear somewhat meaningless within the context of ERFaci without any uncertainty range information.

[Figure]

5. Does the preindustrial simulation include a preindustrial land cover map? A few recent studies show a substantial net decrease in BVOC emissions between preindustrial and present day due to the historical cropland expansion (e.g. Heald et al., 2016; Unger, 2013). Temperate zone forests and grasses have been replaced with crops and pasture that represents a loss of BVOCs from the Earth system. The PI-PD SOA and cloud changes are sensitive to the BVOC emission changes. How will the results be affected in the case of higher PI BVOC emissions? In turn, the higher PI BVOC emissions will influence oxidant levels (reducing them further?). It is unlikely that the higher PI BVOC emissions were included in the oxidant simulations in Lamarque et al., 2010.

6. $\Delta$clean from Ghan (2013) is introduced on Page 4. Readers from gas-phase chemistry community may appreciate a bit more explanation here (1-2 sentences) on the meaning of $\Delta$clean.

---

## Referee Comment (RC3) · Anonymous Referee #2 · 26 Feb 2018

Karset *et al.* explore the importance of using oxidants appropriate to the time period being simulated when determining the strength of the first aerosol indirect effect. Using CAM5.3-Oslo (which includes OsloAero) the authors perform a series of sensitivity simulations to enable them to determine the role of variability for each oxidant individually, and in combination. The manuscript highlights an important issue, is well written throughout, and is both clear and thorough – I would recommend publication in ACP following clarification on the below issues.

**General points**:

It would be useful to see a bit more discussion on the extent to which we can *know* what pre-industrial oxidant levels actually were (or the challenges in determining them). The authors point out that a number of studies have attempted to infer pre-industrial

oxidant levels from emissions inventories (also uncertain) and limited observations, but a bit more detail here about what is known and how well would be useful context.

Relating to my previous point - since there can only be limited confidence in any simulated pre-industrial oxidant levels, one possibility would be to consider how well the full-chemistry model captures present-day oxidant concentrations in clean v. polluted regions. Without this, the paper makes an important point about the potential impact of incorrect / inappropriate oxidant concentrations when diagnosing RFs, but doesn't necessarily tell us how much we can trust this particular set of pre-industrial oxidants and therefore the size of the change in RF that is diagnosed.

Could you add more detail on the new particle formation mechanism that is used in OsloAero? This is also an important factor in determining PI particle concentrations, and therefore the PI to PD radiative forcing. For example, Gordon *et al.*, (2016) found that including pure biogenic new particle formation reduced the strength of their simulated PI to PD first aerosol indirect RF.

**Minor / technical remarks**:

p6, line 29: change to "low volatility"

p7, line 21: change to "describes"

p7, line 26: change to "increases"

p7, line 28: correct the spelling of "switching"

p8, line 14-15: rephrase the sentence starting "Figure 9 . . . . . .", may require insertion of an "is" somewhere?

p8, line 17: should this just be Figure 9(a)? (since (c) and (d) do not relate to DMS?)

p8, line 18-20: I think here you are saying that reaction R2 + R3 is favoured over R4 since there is less oxidation via the NO3 pathway – rephrase to avoid saying "..out off a DMS-"

p11, line 9: change to "gives"

Figure 8: what is meant by "aerosol size" in this Figure? Would be useful to describe in the caption

Figure 9: rephrase third sentence of the caption

Table 5: is there an error here in the description of the NOSOA simulation? (the reaction is the same as the line above)

**References**:

Gordon H., *et al.*, (2016) Reduced anthropogenic aerosol radiative forcing caused by biogenic new particle formation, Proceedings of the National Academy of Sciences of the United States of America, 113, pp.12053-12058. doi: 10.1073/pnas.1602360113.

---

## Referee Comment (RC4) · Anonymous Referee #4 · 3 Mar 2018

This study examines how estimates of the aerosol indirect effect are affected by assumptions regarding preindustrial oxidant levels. The paper is interesting and well executed. For example, in addition to examining the overall effect of preindustrial oxidants, simulations were performed in which present day OH, nitrate, and ozone were replaced with preindustrial levels one at a time. I have just one major comment.

The authors note that the major driver leading to brighter clouds in the preindustrial period compared to default oxidant assumptions is the nitrate radical level. Since the nitrate radical is most abundant at night while daytime oxidation is dominated by OH and ozone, have the authors thought about their results in the context of this diurnal shift towards daytime oxidation? Is the change in AIE mostly due to changes in daytime aerosols? What does that mean if the major oxidant driving changes is primarily

nocturnal? Is nitrate radical oxidation most confined to the lowest model layers as a result of nocturnally stable boundary layers? Thus do the changes in vertical profiles of monoterpenes and DMS in figure 5 mostly reflect an increase in near-surface nocturnal concentrations with PI oxidants? Do any results (Table 3 compound lifetimes?) need to be presented as a daytime average instead of 24-hour average?

Minor comment: Page 8, line 20: "SO2 nucleates easier than SOA" Is SO2 the model species that nucleates due to logistical reasons or is it sulfuric acid?

---

## Author Response (AR1)

**Response to the Referees**

We thank the four reviewers for all the comments on the manuscript. A response to all of the comments are found in the following sections. The comments from the Referees are written in a grey, italic font, followed by our response written in black. The page and line numbers for the changes refers to the revised manuscript with track changes in the end of this document. A complete list of references is also found in this revised manuscript.

**Response to Referee #1**

*"Diurnal Cycle of NO3. The paper identifies the increase in NO3 between the preindustrial and the present day as the major change in oxidants. This is most likely true. However, their model treats NO3 in a relatively unsophisticated manner. The text explicitly says that OH and HO2 concentrations have a diurnal cycle imposed on them whereas NO3 is not mentioned suggesting that it does not. In reality NO3 does have a significant diurnal cycle. Its rapid photolysis leads to low concentrations during the day and high during the night. Thus, in the real atmosphere NO3 anti-correlates with isoprene and monoterpenes (this depends to an extent upon the monoterpene speciation) thus NO3 likely only plays a minor role in the oxidation of these species. By not having a diurnal cycle in the NO3 concentration this anti-correlation is lost in the model which then likely favours the NO3 oxidation route over the other oxidations routes. Given the central importance of NO3 to the primary conclusion of the paper this lack of anti-correlation provides significant problem that needs to be resolved before the paper can go forwards to publication."*

It is correct that our model does not have a diurnal cycle for NO3, and we agree that it should. The diurnal cycles for OH and HO2 were already implemented in the model when this study was started up. The lack of a diurnal cycle for NO3 was discovered after the main simulations for this study were carried out. As a solution to this issue, we have added an extra sensitivity test (from p.13, l.24) where a diurnal cycle to NO3 is applied. The result of this test shows that the lack of this diurnal cycle in the original setup only slightly impacts the result of this study, so we are keeping the original default setup in this study as it is. The main reason for the minor impact of the diurnal cycle of NO3 is that the main effect of NO3 is oxidation of DMS over the oceans. Since the lifetime of DMS is 36 and 55 hours (present-day and preindustrial respectively), the reduction in the nighttime oxidation when not applying a diurnal cycle will have time to be compensated by an increase in the daytime oxidation.

*"Attribution of impacts The chemistry scheme in the model is fundamental to the magnitude of the impacts simulated. There is little explanation for the choice of yield from the VOCs or DMS. A section explaining these choices and previous work to justify these choices would be useful."*

We agree. More information about the yields applied in CAM5.3-Oslo are now included in the model description of this manuscript (p.5, l.12-19). We acknowledge the fact that the yields are uncertain. For both monoterpene and isoprene, the yields are within the range found in other global models and from laboratory experiments (Kroll et al., 2005; Lee et al., 2006; Dentener et al., 2006; Spracklen et al., 2011; Tsigaridis, 2014, Jokinen et al., 2015). Other global models with simplified tropospheric chemistry often do not trace MSA (Neale et al., 2012; Tsigaridis et al., 2014), assuming a yield factor of 0, but since several studies have shown that MSA can contribute to aerosol formation and growth, CAM5.3-Oslo keeps MSA as SOA_LV and SOA_SV in the model after oxidation (Bork et al., 2014; Willis et al., 2016; Chen and Finlayson-Pitts, 2017). The choice of the yields and the ratio between the two are unknown. For that reason, we emphasize the importance of the models representation of DMS-oxidation by the two sensitivity tests in the end of the paper (p.12, l.21-34) where we see how the result is affected if we assume that

1. none of the MSA can contribute to nucleation (NOSOALVDMS, which corresponds to a yield factor for SOA_SV of 100 % and a yield factor for SOA_LV of 0 %)
2. none of the MSA can contribute to either nucleation nor condensation (NOSOA, which corresponds to a yield factor of 0 % for both SOA_SV and SOA_LV).

As shown, the magnitude of the effect on the total aerosol indirect effect by changing the oxidant level is highly dependent on getting some SOA out of the DMS-oxidation reaction that can condense (NOSOA), but it is not that dependent on SOA that can nucleate (NOSOALV). This is specified in the section with the results from the sensitivity tests and in the conclusions (p.15, l.6-8). As part of another research project (BACCHUS - impact of Biogenic versus Anthropogenic emissions on Clouds and Climate: Towards a Holistic UnderStanding), the model applied in this study is taking part in an intercomparison where the choice of yields for the oxidation reactions involving BVOC is studied through several sensitivity tests, but this is beyond the scope of this study.

*"There are couple of ways that the change in oxidants could impact the oxidation of the precursor compounds, and so the production of low volatile products. They could change the net yield of compound (SOA.SV vs SOA.LV), or they could change the location of the oxidation. It would be useful to know which process is occurring here. Given that the yield of SOA.SV is the same for all isoprene oxidation routes, any change in the cloud-aerosol feedback due to isoprene can only be occurring because of the change in the location of oxidation - the globally total production of SOA.SV from isoprene is the same in all simulations (0.15 \* total isoprene emission). Monoterpenes behave differently. Oxidation by O3 leads to the production of SOA.LV whereas oxidation by other methods leads to SOA.SV. Here a change in the oxidant may lead to significant changes in the production of the different SOA times. A similar change might occur with DMS oxidation. The authors attribute all of the changes they see to changes in the location of the oxidation, but they don't really give much evidence to support this.*

We agree that both changes in the location of the oxidation and changes in the net yield of the compounds can contribute to the results in this study. Since the sensitivity tests NOSOALVDMS and NOSOALVBVOC (p.12, l.21-34) show that a shift towards more production of low volatile SOA has a negligible effect on the main result in this study, the attention has been on the changes in the location of the oxidation. With that said, it was not our intention to attribute all of the changes we see to this location change. In fact, the results from the sensitivity test NOSOA (p.13, l.1-14) show a shift towards more production of condensate relative to the production of new particles, which makes it easier for the droplets to activate, also has a big impact on the result. This was highlighted in the section containing the summary and conclusions (p.15, l.6-8). When it comes to the section about the increase in the aerosol number concentration, we agree that the change in the net yield of the compounds could be highlighted more. We have now rewritten this section in an attempt to emphasize this effect (from p.8, l.4 to p.10, l.2), including a table (Table 5) showing how the conversion rates for all the oxidation reactions change when we change the oxidant level.

*Figure 9 shows the change in the fractional oxidation as a function of height, but this doesn't show in absolute terms how much of the precursor is oxidized in different places. This is a little misleading - very little of the monoterpene / isoprene is oxidised in the upper troposphere compared to the lower troposphere. Perhaps showing the absolute change in oxidation would help with this? This would probably need to be on a log scale."*

We agree that most of the precursor gases are oxidized in the boundary layer. To emphasize this, we have added another panel to each part of Fig. 7 (corresponding to Fig. 9 in the old manuscript) showing how much of the specie is oxidized at each level compared to the level where most of the specie is oxidized.

*"It would be useful to have a table or figure which shows how much H2SO4, SOA.LV, and SOA.SV is produced globally by each route for each simulation."*

We agree. This is now included in Table 5 and discussed in the section about the aerosol number concentration (Sect. 4.11, from p.8, l.4 to p.10, l.2).

*"Given the length of the description of the impact on the aerosol and clouds there should be an increase in the description of how the chemistry is making this impact. This explanation of why the emissions are changing the SOA.LV, SOA.SV and H2SO4 tracers is somewhat weak"*

We hope that the inclusion of Table 1 and Table 5, the different sensitivity tests (including the new test with a diurnal cycle for NO3), the extended discussion in Sect. 4.1.1 and the inclusion of Fig. 7 and the corresponding text is satisfactory.

*"Can the total mass of isoprene and monoterpene be included into a table somewhere? The MEGAN scheme can lead to significantly varying emissions depending upon the implementation into the transport model. It would be useful to know these values."*

We agree. It's now included in Table 1.

*"It would also be useful to state that the assumption all of the mono-terpene emission is considered to be alpha-pinene which seems to be the implication of the rate constants chosen."*

Yes, it is correct that all of the monoterpene is considered to be alpha-pinene. This is now added to the manuscript (p.5, l.16).

*"The chemistry scheme also doesn't seem to include aqueous SO2+H2O2. Although this is occurring within the cloud phase it is still a chemical reaction and for completeness I think it should be included."*

Yes, it does. This is specified in the manuscript (p.5, l.16). To what extent the aqueous phase oxidation reactions change with the switch from PD- to PI-oxidants can also be seen in Fig. 7(b).

*"Figure 9: Figure Caption. The language in the figure caption should be re-examined as it doesn't make sense"*

We agree. A new caption is added. (This figure is now called Fig. 7).

**Response to Referee #2**

*"It would be useful to see a bit more discussion on the extent to which we can know what pre-industrial oxidant levels actually were (or the challenges in determining them). The authors point out that a number of studies have attempted to infer pre-industrial oxidant levels from emissions inventories (also uncertain) and limited observations, but a bit more detail here about what is known and how well would be useful context."*

Limited observations refers to simple measurements of surface ozone from a few European stations (Montsouris and Pic DuMidi) in the late 19th century (Volz and Kley, 1988). We have added a sentence about how well the prescribed O3 used in this study corresponds to measured O3 at the Montsouris station in Volz and Kley (1988) (p.5, l.28-30). For the very reactive radicals NO3 and OH, there are no historical measurements. Thus, the PI levels have to be estimated by

models. However, we can have some confidence in these models when they are tested against current observations (e.g. as in Khan et al., for NO3).
We have added a few sentences about the comparison of model simulated concentrations with PD-observations in Sect. 2.2 (p.5, l.31-35).

*"Relating to my previous point - since there can only be limited confidence in any simulated pre-industrial oxidant levels, one possibility would be to consider how well the full-chemistry model captures present-day oxidant concentrations in clean v. polluted regions. Without this, the paper makes an important point about the potential impact of incorrect / inappropriate oxidant concentrations when diagnosing RFs, but doesn't necessarily tell us how much we can trust this particular set of pre-industrial oxidants and therefore the size of the change in RF that is diagnosed."*

We agree with the reviewer that: "Without this, the paper makes an important point about the potential impact of incorrect / inappropriate oxidant concentrations when diagnosing radiative forcings, but does not necessarily tell us how much we can trust this particular set of pre-industrial oxidants and therefore the size of the change in RF that is diagnosed."

The key process identified in our work is the change in oxidation of DMS by NO3. There are a few observations of NO3 from marine sites (Brown and Stutz, 2012). However, most if not all of these are from coastal regions and often in the outflow of major industrialized regions in the northern hemisphere. Thus, measurements of NO3 from the more remote Pacific region that we find is the most important is lacking. On the positive side, the lack of knowledge of PI-levels of NO3 in these regions are probably not very important since they were anyhow very low. Thus, a key factor would be to go out and measure the current levels during the season of DMS production.

When it comes to the other oxidants, we have added a few sentences in Sect. 2.2 (p.5, l.31-35) about how good agreement it is between observations from PD and modeled PD-values of OH and O3 by a full-chemistry model that is almost identical to the model that produced the oxidant fields applied in this study.

*"Could you add more detail on the new particle formation mechanism that is used in OsloAero? This is also an important factor in determining PI particle concentrations, and therefore the PI to PD radiative forcing. For example, Gordon et al., (2016) found that including pure biogenic new particle formation reduced the strength of their simulated PI to PD first aerosol indirect RF."*

We agree that this is an important factor. More information about the new particle formation is added to the model description in Sect. 2 (p.4, l.19-24). We have also added a few sentences

about the lack of pure biogenic new particle formation in the section containing the summary and conclusions, Sect.5 (p.14, l.23-24).

*"Minor / technical remarks:*
*p6, line 29: change to "low volatility""*
Done

*"p7, line 21: change to "describes" p7, "*
Done

*"line 26: change to "increases" p7, "*
Done

*"line 28: correct the spelling of "switching" "*
Done

*"p8, line 14-15: rephrase the sentence starting "Figure 9 . . .. . .", may require insertion of an "is" somewhere?"*
Done

*"p8, line 17: should this just be Figure 9(a)? (since (c) and (d) do not relate to DMS?)"*
Done

*"p8, line 18-20: I think here you are saying that reaction R2 + R3 is favoured over R4 since there is less oxidation via the NO3 pathway – rephrase to avoid saying "..out off a DMS-""*
Done

*"p11, line 9: change to "gives""*
Not changed. "The results give".

*"Figure 8: what is meant by "aerosol size" in this Figure? Would be useful to describe in the caption"*

This was described in the text: "the mean size of the aerosols is calculated as a mean of the number mean radius of all mixtures in the model, weighted by the number of aerosols in each mixture". The explanation is now moved to the caption of the figure instead (Now called Fig. 9).

Done. (Now called Fig. 7)

*"Table 5: is there an error here in the description of the NOSOA simulation? (the reaction is the same as the line above)"*

Yes. That was an error. This is now corrected. (Now called Table 7)

**Response to Referee #3**

*1. The preindustrial oxidant fields are from Lamarque et al., 2010. Is it correct to assume that these oxidant fields were generated without on-line aerosol-cloud radiative interactions i.e. this preindustrial gas-phase chemistry does not "see" the brighter preindustrial clouds found in the present study with CAM5.3-Oslo? This question has broader implications. The changes in aerosol-cloud interactions and associated meteorology in PI versus PD state will have an influence on the resultant oxidant levels, not least through altering photolysis rates. How does the application of off-line oxidants here versus fully 2-way coupled on-line oxidants affect the main results?*

Yes, it is correct that these fields have been generated with a model without online aerosol-cloud radiative interactions, so the gas-phase chemistry is not affected by the brighter PI-clouds we get in this study. We have now pointed out this problem in the summary and conclusion part of the manuscript in Sect. 5 (p.15, l.25-29). This study is the first to highlight the importance of correct treatment of gas-aerosol interactions through oxidation when modeling aerosol indirect effects and the first to address the issue of applying the same oxidant level to both preindustrial and present-day precursor gases. We think that including second-order effects linked up to two-way coupled simulations should be a focus for upcoming studies, as well as applying different kind of input datasets for the oxidants generated by more advanced models, but that this is beyond the scope of this paper.

*2. It is not clear how long the simulations are run for in total?*

The nudged simulations are run for four years, but they are started from simulations with free meteorology using the same oxidant concentrations and the same aerosol concentrations. We have tried to specify this better in the manuscript (p.6, l.19-23).

*However, it is reported that the last 3 years of the run are used for the analyses. ERF allows all feedbacks between land-atmosphere and the land-atmosphere system to come into steady-state*

*with the imposed radiative perturbation. Is the land-atmosphere system in steady-state after only 3 years of running the model? Many of the global chemistry-climate model frameworks seem to run for much longer (even with fixed SSTs and sea ice) to allow for the land-atmosphere system to come into steady-state i.e. more than 20 years.*

It is correct that only the last three years of the four years simulations are analyzed. The standard error for the total aerosol indirect effect for these three years is only 0.01 W/m². Sensitivity simulations (not shown in this paper) show that modeling the total aerosol indirect effect using nudging for 11 years, analyzing the last ten, gives the same result as only running for four years and analyzing the last three, and there is no drift in the signal. Since producing and storing six hourly meteorological data with three dimensions (pressure level, latitude, longitude) and running all the sensitivity tests in this study is computationally expensive, we've chosen to not extend the simulations. This information is now included in the manuscript (p.6, l.23-27). For even more complementary answer to the question about steady-state: see the end of the answer to the next comment.

*3. Is it methodologically correct to 'nudge' a simulation and calculate ERF?*

According to the definition of ERF in IPCC AR5, we acknowledge that nudging the winds is not a 100 % correct method to calculate ERF, since the fast feedbacks on the winds due to aerosol changes are missing and since we do not conserve momentum when replacing ~8 % of the wind signal (similar to a relaxation time scale of six hours) each timestep by a prescribed wind field from another simulation. Since this study consists of several model setups, all with three different simulations each, the computational cost of running all of them with free meteorology will be too high. Since this study is the first to address the issue of using PD-oxidants for both simulations, we think that the tests where we look at the effect of changing one oxidant at a time and the rest of the sensitivity tests are all important for examining the impact of historical oxidant changes on the PD-PI aerosol indirect effect. This would not be affordable for us to do without the use of nudging. Nevertheless, we acknowledge that this issue should be addressed in the paper, so we have added one extra sensitivity test where we run the three simulations from the original default model setup (PDAERO+PDOXI, PIAERO+PDOXI and PIAERO+PIOXI) all over again, but with free meteorology for 50 years each (p.14, l.6-21). Even after 50 years, the uncertainty due to natural variability is large, but the resulting change in the total aerosol indirect effect due to historical oxidant changes falls in the same range for the nudged configuration as for the free running, indicating that the error in the modeled ERF due to nudging is not changing the main results. Related to the previous comment about steady-state, analyzing only the last 30 years of the simulations with free meteorology gives the same result for the total aerosol indirect effect and for the change due to historical oxidant changes as when analyzing the last 50 years, indicating that there is no drift in the signal. This is also pointed out in the manuscript (p.14, l.20-21).

*4. Is it possible to use the 3 model run years to generate a standard error estimate of uncertainty based on interannual internal variability – thus not providing naked numbers e.g. -1.32 W/m2 and -1.07 W/m2. The numbers may appear somewhat meaningless within the context of ERFaci without any uncertainty range information.*

The standard error of the three years with nudging is only 0.01 W/m². We think that adding this uncertainty to the results in this study might give an impression for the readers that the uncertainty linked to historical oxidant changes are low, which is not the case. The uncertainties are high, not only due to interannual variability, but mainly due to other factors that are discussed in the paper (lack of information about PI-oxidant concentrations, simplified model chemistry, uncertainties in yields and more). But we have added standard errors to the modeled total aerosol indirect effect from the new sensitivity test "FREEMET" so the readers can see the magnitude of the interannual variability (p.14, l.17).

*5. Does the preindustrial simulation include a preindustrial land cover map? A few recent studies show a substantial net decrease in BVOC emissions between preindustrial and present day due to the historical cropland expansion (e.g. Heald et al., 2016; Unger, 2013). Temperate zone forests and grasses have been replaced with crops and pasture that represents a loss of BVOCs from the Earth system. The PI-PD SOA and cloud changes are sensitive to the BVOC emission changes. How will the results be affected in the case of higher PI BVOC emissions?*

No, all of the simulations are run with the same land cover map from PD to avoid the result to be impacted by different surface albedos and heat fluxes between the simulations performed with PD- and PI-aerosol emissions. We agree that this also results in almost identical emissions of BVOC (and dust) in the two eras. We are currently preparing another manuscript based on a study of how different kinds of natural emissions have changed since PI due to anthropogenic activities, and how this affects the PD-PI aerosol indirect effect. Preliminary results show that the higher BVOC-emissions in the PI-era due to changes in land cover results in less negative total aerosol indirect effect of ~0.10 W/m². We have yet to test how these emission changes will affect the PD-PI aerosol indirect effect when applying PI-oxidants to the PI-simulation. The lack of differences in many kind of emissions (including BVOC) between the two eras are already mentioned in the summary and conclusion part of the manuscript (from p.15, l.33 to p.16, l.4).

*In turn, the higher PI BVOC emissions will influence oxidant levels (reducing them further?). It is unlikely that the higher PI BVOC emissions were included in the oxidant simulations in Lamarque et al., 2010.*

Yes, we agree that higher BVOC-emissions in PI probably will results in reduced concentrations of the PI-oxidants, especially in the high BVOC-emission regions. The simulations in Lamarque et al. (2010) include prescribed BVOC-emissions, but they are the same for both eras. More recently

developed model versions that will be applied in CMIP6 (like NorESM2, based on CAM6-Oslo) will have prescribed oxidant concentrations produced by a more developed model that probably will see different BVOC-emissions in the two eras due to different land cover maps, and also include online aerosol-cloud radiative interactions (as mentioned in the previous comment). This model version of CAM-Oslo is not yet finalized and ready to be used in this study. It will be interesting to carry out some of the simulations in this study all over again with the CMIP6-models to see how the more developed models will impact the result, but this is beyond the scope of this paper.

*6. Δclean from Ghan (2013) is introduced on Page 4. Readers from gas-phase chemistry community may appreciate a bit more explanation here (1-2 sentences) on the meaning of Δclean.*

We agree. Some extended information about this term is added to the model description (p.4, l.29-31).

**Response to Referee #4**

*The authors note that the major driver leading to brighter clouds in the preindustrial period compared to default oxidant assumptions is the nitrate radical level. Since the nitrate radical is most abundant at night while daytime oxidation is dominated by OH and ozone, have the authors thought about their results in the context of this diurnal shift towards daytime oxidation? Is the change in AIE mostly due to changes in daytime aerosols? What does that mean if the major oxidant driving changes is primarily nocturnal? Is nitrate radical oxidation most confined to the lowest model layers as a result of nocturnally stable boundary layers? Thus do the changes in vertical profiles of monoterpenes and DMS in figure 5 mostly reflect an increase in near-surface nocturnal concentrations with PI oxidants? Do any results (Table 3 compound lifetimes?) need to be presented as a daytime average instead of 24-hour average?*

Since Fig. 3 shows that the changes in the total aerosol indirect effect mainly are results of the effect on the shortwave radiation, it is the daytime aerosol concentration and the daytime cloud properties that are most important. As replied to Referee #1, the model applied in this study unfortunately does not include a diurnal cycle for NO3. This means that the oxidation capacity of NO3 is the same both day and night, and that the oxidation rates will only vary due to the diurnal cycle of some of the emissions (BVOC, but not DMS and SO2). We have added a sensitivity test, DIURNALNO3 (from p.13, l.23), which shows that the main result of this study is not affected by adding a diurnal cycle to the NO3-concentrations. To be able to present different quantities like concentrations, lifetimes and reaction rates as daytime and nighttime averages, all simulations would have to be redone with more frequent output (we only have monthly mean output of these quantities at the moment). Based on the results from the DIURNALNO3-test, we do not think that separate daytime and night-time averages would offer any additional insights important enough to justify the extra work, cost and time that they would require.

*Minor comment: Page 8, line 20: "SO2 nucleates easier than SOA" Is SO2 the model species that nucleates due to logistical reasons or is it sulfuric acid?*

It should be H2SO4, not SO2. This is now corrected (p.10, l.26).

[revised manuscript text omitted]